# SSGNN: SIMPLE YET EFFECTIVE SPECTRAL GRAPH NEURAL NETWORK

## ABSTRACT

Spectral GNNs leverage graph spectral properties to model graph representations but have been less explored due to their computational challenges, especially compared to the more flexible and scalable spatial GNNs, which have seen broader adoption. However, spatial methods cannot fully exploit the rich information in graph spectra. Current Spectral GNNs, relying on fixed-order polynomials, use scalar-to-scalar filters applied uniformly across eigenvalues, failing to capture key spectral shifts and signal propagation dynamics. Though set-to-set filters can capture spectral complexity, methods that employ them frequently rely on Transformers, which add considerable computational burden. Our analysis indicates that applying Transformers to these filters provides minimal advantage in the spectral domain. We demonstrate that effective spectral filtering can be achieved without the need for transformers, offering a more efficient and spectrum-aware alternative. To this end, we propose a *Simple Yet Effective Spectral Graph Neural Network* (SSGNN), which leverages the graph spectrum to adaptively filter through a simplified *global context filtering* approach that captures key spectral features. Moreover, we introduce a novel, parameter-free *Relative Gaussian Amplifier* (ReGA) module, which adaptively learns spectral filtering while maintaining robustness against structural perturbations, ensuring stability. Extensive experiments on 20 real-world graph datasets, spanning both node-level and graph-level tasks along with a synthetic graph dataset, show that SSGNN matches or surpasses the performance of state-of-the-art (SOTA) spectral-based GNNs and graph transformers while using significantly fewer parameters and GFLOPs. Specifically, SSGNN achieves performance comparable to the current SOTA Graph Transformer model, Polynormer, with an average 55x reduction in parameters and 100x reduction in GFLOPs across all datasets. Our code will be made public upon acceptance.

## 1 INTRODUCTION

In recent years, Graph Neural Networks (GNNs) (Scarselli et al. (2008)), have gained significant popularity for machine learning on graph structured data, delivering impressive results on various graph related tasks. Although traditional GNNs utilize a message-passing framework (Gilmer et al. (2017), Battaglia et al. (2018)) to facilitate information exchange between neighboring nodes, they often face challenges such as over-smoothing and over-squashing (Oono & Suzuki (2019), Alon & Yahav (2020), Di Giovanni et al. (2023)) which can restrict their ability to accurately model complex functions. Spectral GNNs (Defferrard et al. (2016), Bruna et al. (2013)), on the other hand, capitalize on graph convolutions and operate in the spectral domain to obtain graph filter responses that enable them to capture non-local dependencies more effectively. Despite the success of spatial GNNs, the exploration of spectral GNNs has been limited, largely because of the use of scalar filters that fail to leverage the rich information within the graph spectrum Bo et al. (2023a). Recently graph transformer (GT) models have show potential for enhancing GNN expressivity, their scalability is limited due to high parameter complexity because of which linear GTs (Choromanski et al. (2022), Zhang et al. (2022), Kong et al. (2023)) have been proposed. However, GTs underperform on numerous widely used datasets (Platonov et al. (2023)), as highlighted in Polynormer (Deng et al. (2024)), which raises concerns about the effective utilization of the expressivity enabled by the self-attention module in these GTs.

Conventional spectral graph filters often apply scalar functions uniformly across eigenvalues, disre-

garding the unique structural insights they provide. It is well known that eigenvalues and eigenvectors capture critical structural properties: smaller eigenvalues represent smooth variations in a graph, while the number of zero eigenvalues reflects the number of connected components. The eigenvector associated with the smaller non-zero eigenvalue balances smoothness and variation across the graph, revealing its global structure. It is common that eigen-values of the graph Laplacian are not distinct. While polynomial filters enhance flexibility by approximating spectral filters with fixed-order polynomials and avoiding costly eigen-decomposition, they struggle with the high multiplicity of eigenvalues commonly found in real-world graphs. This multiplicity leads to uniform scaling of frequency components with the same eigenvalue, limiting the expressive capacity of these filters.

Unlike traditional scalar-to-scalar filters that apply the same filter to all eigenvalues, set-to-set filtering leverages the spectrum information from a set of eigenvalues, enabling more expressive spectral filtering by modeling complex interactions between frequency components. Specformer (Bo et al. (2023a)) employs transformers (Vaswani et al. (2017)) for this purpose, directly capturing eigenvalue relationships at the graph spectrum level without relying on polynomial approximations, thereby constructing advanced spectral filters. However, its self-attention mechanism introduces a significant computational bottleneck with quadratic time and space complexity, limiting scalability to large graphs (Bo et al. (2023b)) and causing CUDA OOM issues even on moderately sized heterophilic datasets. Heterophilic graphs, where nodes from different classes tend to connect, are highly sensitive to hyperparameters (Luan et al. (2024)). Transformers also exhibit hyperparameter sensitivity and training instability due to parameter perturbations (Chen et al. (2021)). Based on these challenges, we investigate: *"Can we develop an effective spectral filtering approach that adaptively learns filters based on graph data characteristics in a parameter and computation efficient manner, while capturing global eigen-spectrum?"*

We introduce SSGNN: *Simple Yet Effective Spectral Graph Neural Network*, an encoder-decoder architecture that directly operates on the graph spectrum. Designed for computational efficiency, SSGNN incorporates a novel, parameter-free *Relative Gaussian Amplifier* (ReGA) module, which enables dynamic spectral filtering via *global context*, capturing *inter-eigenvalue relationships* without the computational overhead of quadratic time complexity, making it scalable for large graphs.

This lightweight design enables SSGNN to effectively model a diverse range of homophilic and heterophilic datasets. Our results demonstrate that SSGNN not only achieves comparable but often surpasses the performance of state-of-the-art (SOTA) graph transformers (GT) and spectral GNNs while significantly reducing parameter counts. Figure 1 provides a comparative analysis of SS-GNN with Specformer and Polynormer Deng et al. (2024), the current SOTA spectral GNN and GT on WikiCS (homophilic graph) and Squirrel (heterophilic graph) datasets. SSGNN achieves the highest accuracy on both datasets while being fast and minimizing parameter usage.

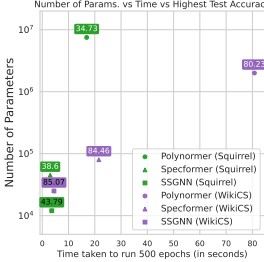

Figure 1: Comparison of SSGNN with Specformer and Polynormer.

The major contributions of the paper can be summarised as follows:

- We propose a simple architecture featuring: i) a spectral encoder to capture eigenvalue interdependencies and global structural information, while incorporating eigen-correction; ii) a decoder that serves as a bank of filter bases, enabling adaptive learning of spectral filters tailored to the graph's characteristics, enhanced by our parameter-free **ReGA** module, which introduces global context filtering.

- We provide both theoretical and empirical analysis showing that SSGNN is robust against structural changes.

- Experiments on a synthetic dataset demonstrate that SSGNN can learn complex filters, providing accurate approximations of spectral filters compared to other spectral GNNs and GT. Specifically, SSGNN excels in modeling high-pass, band-pass, and comb filters, achieving performance nearly 100 times better than Specformer in node regression task.

- Extensive experiments across 20 real-world graph datasets for node and graph classification tasks demonstrate that SSGNN achieves comparable or even surpasses recent SOTA models, with nearly 100x fewer parameters. Specifically, for node classification, on the

homophilic WikiCS dataset, SSGNN outperforms Polynormer by **5%**, utilizing approximately **2700x** fewer parameters and **2600x** fewer GFLOPs. In the same vein, on the heterophilic Tolokers dataset, SSGNN achieves nearly 2% higher AUC compared to Polynormer, with about **2600x** fewer parameters and **500x** fewer GFLOPs. In graph classification tasks, we achieve a new best mean absolute error (MAE) of 0.0592 on the ZINC dataset and 0.3012 average precision (AP) on MolPCBA. Additionally, on MolHIV, we surpass an AUC of 80, a benchmark previously reached only by higher-order GNNs like CIN (Bodnar et al. (2021)).

## 2 BACKGROUND AND RELATED WORKS

**Preliminary.** Consider an undirected graph $\mathcal{G} = (\mathcal{V}, \mathcal{E})$, where $\mathcal{V}$ represents a finite set of $N$ nodes. Each node $i \in \mathcal{V}$ is associated with a feature vector $\boldsymbol{X}_i \in \mathbb{R}^d$, where $\boldsymbol{X} \in \mathbb{R}^{N \times d_f}$ is the node feature matrix and $d_f$ denotes the dimensionality of node features. $\mathcal{E} \subseteq \mathcal{V} \times \mathcal{V}$ represents the edge set and the adjacency matrix of $\mathcal{G}$ is denoted as $\boldsymbol{A} \in \{0,1\}^{N \times N}$. Let $\boldsymbol{D}$ be the diagonal degree matrix, where $D_{ii}$ corresponds to the degree of the node $i$. $\boldsymbol{L} = \boldsymbol{I}_N - \boldsymbol{D}^{-1/2}\boldsymbol{A}\boldsymbol{D}^{1/2}$ denotes the normalized graph Laplacian, where $\boldsymbol{I}_N$ is the identity matrix. Since $\boldsymbol{L}$ is a real symmetric matrix, its eigen decomposition can be defined as $\boldsymbol{L} = \boldsymbol{U}\boldsymbol{\Lambda}\boldsymbol{U}^T$, where $\boldsymbol{\Lambda} = \text{diag}(\lambda_0, \lambda_1, ..., \lambda_N)$ is the diagonal matrix of the eigenvalues and $\boldsymbol{U} = [u_0, u_1, ..., u_n]$ comprises of the corresponding eigenvectors.

**Graph Signal Processing.** The *graph Fourier transform* (GFT) for a signal $\boldsymbol{x} \in \mathbb{R}^{N \times 1}$ is expressed as $\hat{\boldsymbol{x}} = \boldsymbol{U}\boldsymbol{x} \in \mathbb{R}^{N \times 1}$. A spectral filter $g_\theta$ operates on $\hat{\boldsymbol{x}}$ to scale the Fourier coefficients. Subsequently, an inverse GFT is applied to obtain the filtered signal in the vertex domain as $\boldsymbol{x} = \boldsymbol{U}\hat{\boldsymbol{x}}$.

Existing GNNs can be roughly divided into three categories: Spatial GNNs, Spectral GNNs and GT.

**Spatial GNNs.** Spatial GNNs operate by leveraging local-message passing mechanisms, where information is exchanged between neighboring nodes based on graph topology. GraphSAGE Hamilton et al. (2017) uses efficient sampling and aggregation, MPNN Gilmer et al. (2017) generalizes message-passing across nodes, GAT Velickovic et al. (2017) as well as GATv2 Brody et al. (2021) incorporate attention mechanisms to weigh the importance of neighboring nodes. Although stacking multiple layers in spatial GNNs facilitates the capturing of long-range dependencies, it frequently introduces difficulties such as over-smoothing Oono & Suzuki (2019), making node representations hard to distinguish, and over-squashing Topping et al. (2021), where distant information is compressed into limited node capacity.

**Spectral GNNs.** Spectral GNNs (Wu et al. (2019), Dong et al. (2020)) operate in the frequency domain, utilizing the graph's Laplacian spectrum to perform convolutions and capture global structures. Initial models like Spectral CNN (Bruna et al. (2013)) and ChebNet (Defferrard et al. (2016)) laid the foundation by learning convolutional filters in the spectral domain, while Graph Convolutional Networks (GCN) (Kipf & Welling (2016)) further simplified spectral convolution for broader applicability. Recent advances include SpecFormer (Bo et al. (2023a)), which integrates spectral graph convolution with transformer architectures to capture local and global patterns effectively. $G^2$CN (Li et al. (2022)) utilizes Gaussian convolutional networks with concentrated graph filters, enhancing efficiency while preserving structural properties. Despite these developments, spectral methods still struggle with computational intensity and scalability challenges, making them less practical for very large graphs (Wu et al. (2021), Liang et al. (2022)).

**Graph Transformers.** Graph Transformers (Thekumparampil et al. (2018), Yun et al. (2019)) have emerged as a powerful alternative to traditional GNNs, effectively modeling long-range dependencies through attention mechanisms that capture both local and global structures. Key advancements include Graphormer (Ying et al. (2021)), which embeds structural encodings for improved performance on molecular graphs; SAN (Kreuzer et al. (2021)), which integrates structural relationships to enhance node classification; and GraphGPS (Rampášek et al. (2022)), which combines local message-passing with global attention. Efficiency-focused models like Exformer (Shirzad et al. (2023)) and DiFFormer (Gao et al. (2022)) aim to optimize attention mechanisms for scalability and better long-range dependency modeling.

## 3 METHODOLOGY

In this section, we present the architecture of SSGNN, which includes a Spectral Encoder and a Decoder. Then, we discuss our parameter-free ReGA module followed by an overview of the modified graph convolution. Finally, we conclude with the computational complexity analysis of SSGNN.

### 3.1 SPECTRAL ENCODER

First we adopt the eigen-correction strategy from Lu et al. (2024), to address the issue of repeated eigenvalues as discussed in Section. 1, modifying the eigenvalues as follows:

$$\bar{\lambda}_i = \beta\lambda_i + (1 - \beta)\frac{2i}{N-1}, \forall i \in N, \tag{1}$$

where $\beta \in [0,1]$ is a hyperparameter that ensures $\bar{\lambda}_i$ remains strictly monotonically increasing, making each $\bar{\lambda}_i$ to be unique. Then, we define an encoding function motivated by Specformer Bo et al. (2023a), which maps these correct scalar eigenvalues $\phi : \mathbb{R} \mapsto \mathbb{R}^d$, into meaningful $d$ dimensional vector representation, as:

$$\phi(\bar{\lambda}_k, 2i) = \sin((\epsilon\bar{\lambda}_k)/10000^{2i/d}) \; ; \; \phi(\bar{\lambda}_k, 2i+1) = \cos((\epsilon\bar{\lambda}_k)/10000^{2i/d}), \, \forall k \in N. \tag{2}$$

where $\epsilon > 0$ denotes the scaling factor. While Specformer emphasizes the advantages of encoding for expressive scalar eigenvalue representation, multi-scale features, and relative frequency shifts, we offer a new perspective that supports our subsequent approach. We begin by concatenating the corrected eigenvalues $\bar{\lambda}_i \forall i \in \mathbb{R}^N$ with their corresponding encodings $\phi(\bar{\lambda}_i)$ to form $\boldsymbol{Z}_{eig} = (\bar{\lambda}_0 \| \phi(\bar{\lambda}_0), \dots, \bar{\lambda}_n \| \phi(\bar{\lambda}_n)) \in \mathbb{R}^{N \times (d+1)}$. Next, we apply a transformation $\boldsymbol{W}_{eig} \in \mathbb{R}^{(d+1) \times (d+1)}$ to $\boldsymbol{Z}_{eig}$ to learn the dependencies between the original eigenvalues and their encoded representations, which can be seen as $\hat{\boldsymbol{Z}}_{eig} = \boldsymbol{Z}_{eig}\boldsymbol{W}_{eig}$, $\hat{\boldsymbol{Z}}_{eig} \in \mathbb{R}^{N \times (d+1)}$. $\hat{\boldsymbol{Z}}_{eig}$ can be interpreted as comprising of two key components: (i) the corrected eigenvalues $\bar{\lambda}_i$, which retains essential global structural information and (ii) $\phi(\bar{\lambda}_i)$, which captures the oscillatory behavior of these corrected eigenvalues.

In homophilic data, where nodes with similar attributes are connected, the signal predominantly exhibits low-frequency characteristics. This is due to connected nodes typically sharing comparable features, resulting in a smoother graph signal. Consequently, the variations captured by $\phi(\bar{\lambda})$ are minimal, highlighting the critical role of the corrected eigenvalue in enabling the model to distinguish between encodings. In contrast, heterophilic data, characterized by connections between nodes with dissimilar attributes, is captured by high-frequency components. Here, connected nodes exhibit substantial variability, leading to rapid fluctuations in the graph signal. These rapid oscillations are represented by the higher eigenvalues of the Laplacian, while $\phi(\bar{\lambda})$ encapsulates these dynamic variations within the graph structure. The weight matrix $\boldsymbol{W}_{eig}$ is trained to learn the interdependencies between the corrected eigenvalue and its encoding, facilitating optimal model performance. By leveraging this encoding, the model adapts to the underlying graph structure, accurately capturing both smooth and oscillatory behaviors.

### 3.2 DECODER

The output of the encoder, $\hat{\boldsymbol{Z}}_{eig} \in \mathbb{R}^{N \times (d+1)}$, is then passed into a two-layer MLP-based decoder that includes a non-linear activation function. This architecture enables the decoder to effectively capture intricate transformations and adapt versatile spectral filtering. It is given by:

$$\lambda_h = \sigma(\hat{\boldsymbol{Z}}_{eig}\boldsymbol{W}_1)\boldsymbol{W}_h, \; \lambda_h \in \mathbb{R}^{N \times 1}, \; \boldsymbol{W}_1 \in \mathbb{R}^{(d+1) \times (d+1)}, \; \boldsymbol{W}_h \in \mathbb{R}^{(d+1) \times 1}. \tag{3}$$

Here $\sigma$ denotes activation and $\boldsymbol{W}_1, \boldsymbol{W}_h$ are learnable parameters. The decoder holds a crucial function in our network, being responsible for adaptively executing spectral filtering. Its main objective is to learn a wide variety of filtering strategies by adjusting to the graph's spectral characteristics and the node features. To allow the decoder to capture various aspects of the filtering procedure, we introduce $H$ heads, drawing inspiration from multi-head architecture of Transformers. These heads allow the decoder to learn diverse spectral filtering patterns. These heads focus on learning distinct aspects of the signal, making the decoder a bank of efficient filter bases (Effect of multi-head visualised in

Appendix. F). As training progresses, the encoder refines its representations by integrating node feature data. However, in the early stages, the decoder might face difficulties in grasping meaningful representations due to its reliance on spectral encodings, which may not comprehensively capture all dependencies. Consequently, the learned filters may exhibit uneven amplitude distribution, where a filter intended for a specific frequency band inadvertently captures signals from other bands.

To tackle the problem of noisy focus by decoder on irrelevant features, which hinders its effectiveness in accurately isolating the target frequency band, we implement a mean shift at the decoder level. This adjustment re-centers the amplitude distribution around zero, facilitating a more balanced and accurate filtering mechanism. Furthermore, the variations learnt across different decoder heads enhance the diversity of the filtering process. In this multi-head configuration, the mean shift allows each head to concentrate more effectively on its target frequency components, yielding filters that are both consistent and specialized. The mean shift is defined as $\lambda_h - \mu_h$, $\forall h \in H$, where $\mu_h = \frac{1}{N}\sum_{k=1}^{N} \lambda_{hk}$ denotes the mean amplitude of the filter head. This mean value encapsulates the global context captured by the decoder for that specific head. However, using the mean shift directly may lead to inconsistencies, as the distribution can fluctuate unpredictably due to varying frequency components. To address this, scaling the mean shift by the standard deviation of each head's amplitudes $\sigma_h$ normalizes these fluctuations, resulting in a more balanced representation across all frequency bands. Considering these we introduce a head-specific factor $z_h \in \mathbb{R}^{N \times 1}$ defined by:

$$z_h = \frac{1}{\sigma_h}(\lambda_h - \mu_h) \tag{4}$$

Here $\sigma_h$ represents the standard deviation of the learned amplitudes for each head $h$, calculated as $\sigma_h = \sqrt{\frac{1}{N}\sum_{k=1}^{N}(\lambda_{hk} - \mu_h)^2} + \tilde{\epsilon}$ with $\tilde{\epsilon}$ being a small constant added for numerical stability. We find that this formulation is equivalent to normalizing the learned amplitudes for each head of the decoder. By applying mean shift, we obtain the global context of the entire spectrum. This formulation serves as the foundation for global context filtering. (Appendix G.5 explores the impact of mean shift.)

### 3.3 ReGA: Relative Gaussian Amplifier

To further enhance the representational capabilities of the filters learned by the decoder, we concentrate on amplification at the filtering level. [1] By implementing a function $f(\cdot)$ to adjust the amplification of the decoder's filters, we can optimize their ability to capture relevant spectral features, thereby improving overall efficacy. For homophilic graph data, $f(\cdot)$ should guide the decoder to amplify low-frequency components, allowing the filter to effectively function as a low-pass filter. In contrast, for heterophilic graph data, $f(\cdot)$ should enable the decoder to adaptively model a band-reject filter or amplify high-frequency components, thereby acting as a high-pass filter. $z_h$ highlights a preference for frequency components with higher amplitudes. To emphasize this distinction, we aim to model $f(\cdot)$ in a way that amplifies the amplitudes of frequency components near the mean while suppressing those that diverge from it. This approach is designed to meet the following criteria: **i)**. $f(z)$ is constrained within the semi-closed interval $(0, 1]$, meaning $f(z) \in (0, 1]$; **ii)**. It attains a unique maximum value of 1 at $z = 0$; **iii)**. As $z$ approaches infinity, the function asymptotically converges to zero, represented by $\lim_{z \to \pm\infty} f(z) = 0$. To meet these properties, we propose *Relative Gaussian Amplifier* (ReGA), $G_h(z_h)$ utilizing the Gaussian function over other functions as

$$G_h(z_h) = \alpha e^{-\frac{(z_h - a)^2}{2b^2}}, \ \forall h \in H, \tag{5}$$

where $\alpha$ represents the scale of the Gaussian function. We set $\alpha = 1$ to satisfy the first condition. The mean $a$ is set to 0 to meet the second condition and $b$ is the standard deviation, serving as a hyperparameter controlling the spread of the scores across different amplitude values. Thus, the simplified form of $G_h(z_h)$ becomes:

$$f_{z_h} = G_h(z_h) = e^{-\frac{(z_h)^2}{2b^2}}, \ f_{z_h} \in \mathbb{R}^{N \times 1}, \tag{6}$$

---

[1] In filtering, this entails prioritizing relevant frequency components to enhance their influence while diminishing the effects of less relevant ones. From this point forward, we will refer to this adaptive behavior as amplification throughout the paper.

where $f_{z_h}$ represents ReGA's output, the adaptive scores for each head $h$. $f_{z_h}$ is now applied element-wise to the decoder's output $\lambda_h \in \mathbb{R}^{N \times 1}$, yielding the scaled output $\hat{\lambda}_h$. This operation can be expressed as: $\hat{\lambda}_h = f_{z_h} \odot \lambda_h$ where $\odot$ denotes element-wise multiplication. By integrating ReGA, the decoder effectively adapts its learned filter bases, resulting in improved filtering of frequency components. (Appendix. G.4 explores the impact of ReGA.)

To gauge the amplification effect of the filters on the initial values, we quantify the difference ($\hat{\lambda}_{ih} - \lambda_i, \ \forall i \in \mathbb{R}^N$), highlighting the degree of emphasis placed on each component during filtering (here $\lambda_i$) denotes the original eigenvalue). This approach is inspired by the gradient operator in graph diffusion techniques, as discussed by Chamberlain et al. (2021). Amplified components reflect heightened attention, while suppressed components denote reduced influence, thereby enhancing the model's adaptiveness. We conceptualize this deviation as a learned spectral shift. To improve the interpretability of the learned spectral shift, we introduce a constraint to prevent negative filtering (Bo et al. (2021)). This constraint is crucial for avoiding adverse effects in the spectral domain that could arise from negative amplitude values. We enforce non-negativity by applying the absolute value operation, represented as $\hat{\lambda}_{ih} = |\hat{\lambda}_{ih} - \lambda_i|, \ \forall i \in \mathbb{R}^N$.

## 3.4 GRAPH CONVOLUTION

With the learned spectral filters $\hat{\lambda}_h$, we now move on to defining the spectral bases necessary for graph convolution. For each head $h \in H$, we construct the corresponding spectral bases $\boldsymbol{F}_h$, stack them along the head dimension, and pass these stacked bases through an MLP to capture interdependencies between the spectral bases of different heads. This process is formulated as:

$$\boldsymbol{F}_h = \boldsymbol{U}\text{diag}(\boldsymbol{\lambda}_h)\boldsymbol{U}^\top, \quad \hat{\boldsymbol{F}} = \text{MLP}([\boldsymbol{F}_0||\cdots||\boldsymbol{F}_{H-1}]), \tag{7}$$

where $\hat{\boldsymbol{F}}$ is the final set of learned spectral bases that model interdependencies across heads. These learned bases serve a similar role to polynomial bases used in existing literature, but are adaptively learned from the data. Finally our graph convolution can be defined as:

$$\bar{\boldsymbol{X}}^{(l-1)} = \hat{\boldsymbol{F}}\boldsymbol{X}^{(l-1)}, \ \ \boldsymbol{X}^{(l)} = \sigma(\bar{\boldsymbol{X}}^{(l-1)}\boldsymbol{W}^{(l-1)}), \tag{8}$$

where $\boldsymbol{X}^{(l)}$ represents the node representations for the $l$-th layer, while $\bar{\boldsymbol{X}}^{(l-1)}$ represents the modified representations of $(l-1)$-th layer when given the spectral bases $\hat{\boldsymbol{F}}$. The matrix $\boldsymbol{W}^{(l-1)}$ represents the transformation that updates $\bar{\boldsymbol{X}}^{(l-1)}$ to $\boldsymbol{X}^{(l)}$ and $\sigma$ represents the activation. By stacking multiple graph convolutional layer, SSGNN effectively learns node representations.

## 3.5 THEORITICAL ANALYSIS AND COMPUTATIONAL COMPLEXITY

**Theoritical Analysis**. Let $\lambda \in \mathbb{R}^{N \times 1}$ be the eigenvalue and $\tilde{\lambda} = \lambda + \xi$ represent a pertubed eigenvalue, where $\xi$ is a bounded pertubation such that $\|\xi\|_2 \leq \delta$. The Gaussian-based spectral filtering $f(\lambda) = \exp\left(-\frac{(\Lambda - \mu)^2}{2(\sigma c)^2}\right)$ remains stable under perturbations in the eigenvalue, such that:

$$\|f(\lambda) - f(\hat{\lambda})\|_2 \leq K\delta \tag{9}$$

where $K$ is a constant and $\delta$ is the perturbation bound. All proofs and **empirical** analysis are provided in the Appendix section C

**Computational Complexity**. SSGNN's computation involves two key components: spectral decomposition and forward process. Spectral decomposition, pre-computed with a complexity of $\mathcal{O}(N^3)$, is amortized over multiple training and inference steps, as it is computed just once and stored. For smaller graphs, this precomputation incurs minimal overhead (Appendix. G.1). However, for larger graphs, fast numerical methods such as Krylov subspace approximations for the top $k$ eigenvalues or Sparse Generalized Eigenvalue algorithms can significantly reduce the cost by efficiently estimating $k$ eigenvalues and eigenvectors. The forward pass has two parts: learnable bases and graph convolution, with complexities $\mathcal{O}(HN^2)$ and $\mathcal{O}(lNd)$, respectively, where $l$ represents the number of layers, and $d$ is the hidden dimension. Thus the total forward complexity is $\mathcal{O}(N(HN + ld))$. For large graphs, this complexity can be further reduced to $\mathcal{O}(Hk^2 + lNd)$. (Training time comparison is detailed in Appendix. G.2).

Table 1: Node regression: Mean of the sum of squared error and ($R^2$ score) on synthetic data.

| Model | Low-pass $\exp(-10\lambda^2)$ | High-pass $1 - \exp(-10\lambda^2)$ | Band-pass $\exp(-10(\lambda - 1)^2)$ | Band-rejection $1 - \exp(-10(\lambda - 1)^2)$ | Comb $|\sin(\pi\lambda)|$ |
|---|---|---|---|---|---|
| GCN | 3.4799(.9872) | 67.6635(.2364) | 25.8755(.1148) | 21.0747(.9438) | 50.5120(.2977) |
| GAT | 2.3574(.9905) | 21.9618(.7529) | 14.4326(.4823) | 12.6384(.9652) | 23.1813(.6957) |
| ChebyNet | 0.8220(.9973) | 0.7867(.9903) | 2.2722(.9104) | 2.5296(.9934) | 4.0735(.9447) |
| GPR-GNN | 0.4169(.9984) | 0.0943(.9986) | 3.5121(.8551) | 3.7917(.9905) | 4.6549(.9311) |
| BernNet | 0.0314(.9999) | 0.0113(.9999) | 0.0411(.9984) | 0.9313(.9973) | 0.9982(.9868) |
| JacobiConv | **0.0003(.9999)** | 0.0064(.9999) | 0.0213(.9999) | 0.0156(.9999) | 0.2933(.9995) |
| Specformer | 0.0048(.9999) | 0.001(.9999) | 0.000057(.9999) | **0.0054(.9999)** | 0.0052(.9999) |
| SSGNN | 0.0034(.9999) | **0.000008(.9999)** | **0.0000059 (.9999)** | 0.0084(.9999) | **0.000057(.9999)** |

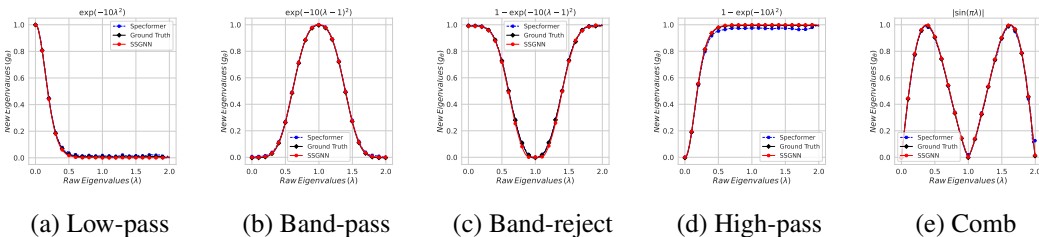

| (a) Low-pass | (b) Band-pass | (c) Band-reject | (d) High-pass | (e) Comb |

Figure 2: Filters learned by Specformer and SSGNN on a Synthetic dataset

## 4 EXPERIMENTS

In this section, we present experiments on diverse real-world and a synthetic graph dataset to assess the effectiveness of our model, SSGNN. We thoroughly compare its performance against SOTA GNNs, Graph Transformer and spectral GNN models across both homophilic and heterophilic graphs for node classification, graph classification and regression. We also perform ablation studies to show the effect $\epsilon$ and $b$ on the learned filters after our ReGA operation.

### 4.1 LEARNING SPECTRAL FILTERS ON SYNTHETIC DATA

In this experiment, we use the synthetic dataset from Specformer, where 50 images from the Image Processing Toolbox are treated as 2D 4-neighbor grid graphs with pixel values as node features. All images share the same adjacency matrix, and five predefined graph filters generate the ground truth signals. Specformer-Small and SSGNN are both configured with 16 hidden units and 1 head for fair comparison, with training running up to 2000 epochs. No regularization is applied, and a learning rate of 0.01 is used. We measure performance using sum of squared error and $R^2$ score. We conduct experiments using Specformer-Small and SSGNN with the described settings, while for the remaining baselines, we rely on the results reported in the Specformer paper, as it already provides a fair comparison across models. Table 1 shows the quantitative results, where SSGNN consistently outperforms other models, especially on complex filters like high-pass, band-pass, and comb achieving lower SSE. SSGNN surpasses Specformer even in low-pass filtering. Figure 2 visualizes the learned filters, showing that SSGNN aligns more closely with ground truth across all scenarios. These results highlight that SSGNN is not only lightweight but also highly effective in capturing various types of filtering based on the data.

### 4.2 NODE CLASSIFICATION

In the node classification task, we assess SSGNN using eight homophilic datasets and nine heterophilic datasets. In Tables 2, 3 and 4; "∗" indicates the baselines that were trained from scratch. Details on the baseline settings and dataset splits can be found in the Appendix. E.

**Results.** The node classification results span three tables, each analyzed in detail. . In Table 2, SSGNN shows competitive performance on homophilic datasets, outperforming most of SOTA GT models. Notably SSGNN achieves a record $85.16\%$ accuracy on WikiCS, surpassing Polynormer by 5% with a remarkable **625x** reduction in parameters.On Coauthor-Physics, Specformer faces OOM, while SSGNN ranks first without such issues. In Amazon-Photo, SSGNN ranks second with comparable performance to Specformer, lower standard deviation, and a **300x** reduction in param-

eters compared to Polynormer. In large-scale graphs like ogbn-arXiv, SSGNN outperforms both Polynormer and Specformer, with Polynormer initially facing OOM issues. In Table 3, we evaluate homophilic and heterophilic datasets targeted by Specformer, which Polynormer does not address. For Chameleon, SSGNN achieves accuracy improvements of 7% and 2.35% over Specformer and Polynormer, respectively. In the Squirrel dataset, it records increases of 4.65% and 7.45% compared to Specformer. These significant enhancements demonstrate SSGNN's superior ability to adaptively capture high-pass and band-rejection features, as further evidenced by the synthetic data experiments (see Table 1). In Table 4, we present results from experiments on the latest heterophilic graph datasets introduced by Platonov et al. (2023), which were evaluated using Polynormer but not by Specformer. We encountered challenges with the Roman-Empire dataset due to its directed nature, which hinders spectral GNNs that convert it to undirected graphs. In contrast, models like Dir-GNN, designed for directed graphs, achieve superior results. In the Minesweeper dataset, we ranked second relative to Polynormer with a notable **3000x** reduction in parameters. In the Tolokers and Questions datasets, we achieve new SOTA accuracies, surpassing Polynormer and Specformer by 1.86% and 1.36% respectively, while also achieving a **2600x** reduction in parameters compared to Polynormer. These results demonstrate that SSGNN not only surpasses SOTA models across diverse graph datasets but also showcases improved scalability and efficiency in handling both homophilic and heterophilic graph data. Its streamlined architecture, combined with effective, parameter-free ReGA module, allows SSGNN to effectively learn complex filters.

Table 2: Results for node classification accuracy on homophilic datasets. We report average accuracy (%) $\pm$ std over 10 runs. The top **first**, **second**, and **third** results are highlighted for each dataset.

| | Computer | Photo | CS | Physics | WikiCS | ogbn-arXiv |
|---|---|---|---|---|---|---|
| GCN | $89.65 \pm 0.52$ | $92.70 \pm 0.20$ | $92.92 \pm 0.12$ | $96.18 \pm 0.07$ | $77.47 \pm 0.85$ | $71.74 \pm 0.29$ |
| GraphSAGE | $91.20 \pm 0.29$ | $94.59 \pm 0.14$ | $93.91 \pm 0.13$ | $96.49 \pm 0.06$ | $74.77 \pm 0.95$ | – |
| GAT | $90.78 \pm 0.13$ | $93.87 \pm 0.11$ | $93.61 \pm 0.14$ | $96.17 \pm 0.08$ | $76.91 \pm 0.82$ | $72.01 \pm 0.20$ |
| GCNII | $91.04 \pm 0.41$ | $94.30 \pm 0.20$ | $92.22 \pm 0.14$ | $95.97 \pm 0.11$ | $78.68 \pm 0.55$ | – |
| GPRGNN | $89.32 \pm 0.29$ | $94.49 \pm 0.14$ | $95.13 \pm 0.09$ | $96.85 \pm 0.08$ | $78.12 \pm 0.23$ | $71.10 \pm 0.12$ |
| OrderedGNN | $92.03 \pm 0.13$ | $95.10 \pm 0.20$ | $95.00 \pm 0.10$ | $97.00 \pm 0.08$ | $79.01 \pm 0.68$ | – |
| GraphGPS | $91.19 \pm 0.54$ | $95.06 \pm 0.13$ | $93.93 \pm 0.12$ | $97.12 \pm 0.19$ | $78.66 \pm 0.49$ | $70.97 \pm 0.41$ |
| Exphormer | $91.47 \pm 0.17$ | $95.35 \pm 0.22$ | $94.93 \pm 0.01$ | $96.89 \pm 0.09$ | $78.54 \pm 0.49$ | $72.44 \pm 0.28$ |
| NodeFormer | $86.98 \pm 0.62$ | $93.46 \pm 0.35$ | $95.64 \pm 0.22$ | $96.45 \pm 0.28$ | $74.73 \pm 0.94$ | $67.19 \pm 0.83$ |
| DIFFormer | $91.99 \pm 0.76$ | $95.10 \pm 0.47$ | $94.78 \pm 0.20$ | $96.60 \pm 0.18$ | $73.46 \pm 0.56$ | $69.86 \pm 0.25$ |
| Specformer | $87.23 \pm 0.52^*$ (33K) | $95.36 \pm 0.32^*$ (32K) | $95.60 \pm 0.07^*$ (226K) | $OOM^*$ | $84.55 \pm 0.20^*$ (17.6K) | $71.98 \pm 0.33^*$ (500K) |
| Polynormer | $93.71 \pm 0.21^*$ (5.4M) | $96.48 \pm 0.34^*$ (7.8M) | $95.55 \pm 0.11^*$ (9.3M) | $97.27 \pm 0.11^*$ (4.0M) | $80.07 \pm 0.56^*$ (7.5M) | $71.89 \pm 0.21^*$ (393K) |
| SSGNN | $91.38 \pm 0.38$ **(27.2K)** | $95.38 \pm 0.03$ **(26.5K)** | $96.30 \pm 0.08$ **(220K)** | $98.33 \pm 0.15$ **(135K)** | $85.16 \pm 0.41$ **(12.3K)** | $72.10 \pm 0.04$ **(36.5K)** |

Table 3: Averaged accuracy (%) $\pm$ std over 10 runs for node classification on homophilic *(Cora, Citeseer)* and heterophilic datasets. We highlight the top **first**, **second**, and **third** results per dataset.

| | Chameleon | Squirrel | Actor | Penn94 | Cora | Citeseer |
|---|---|---|---|---|---|---|
| GCN | $40.89 \pm 4.12$ | $39.47 \pm 1.47$ | $33.23 \pm 1.16$ | $82.47 \pm 0.27$ | $87.14 \pm 1.01$ | $79.86 \pm 0.67$ |
| GAT | $39.21 \pm 3.08$ | $35.62 \pm 2.06$ | $33.93 \pm 2.47$ | $81.53 \pm 0.55$ | $88.03 \pm 0.79$ | $80.52 \pm 0.71$ |
| H$_2$GCN | $26.75 \pm 3.64$ | $35.10 \pm 1.15$ | $35.86 \pm 1.03$ | $OOM$ | $86.92 \pm 1.37$ | $77.07 \pm 1.64$ |
| GPRGNN | $39.93 \pm 3.30$ | $38.95 \pm 1.99$ | $39.92 \pm 0.67$ | $81.38 \pm 0.16$ | $88.57 \pm 0.69$ | $80.12 \pm 0.83$ |
| JacobiConv | $39.00 \pm 4.20$ | $29.71 \pm 1.66$ | $41.17 \pm 0.64$ | $83.35 \pm 0.11$ | $88.98 \pm 0.46$ | $80.78 \pm 0.79$ |
| Specformer | $36.11 \pm 0.44^*$ (82k) | $37.66 \pm 0.42^*$ (75K) | $42.01 \pm 1.14^*$ (37k) | $84.28 \pm 0.32^*$ (338K) | $88.50 \pm 0.98^*$ (54K) | $81.52 \pm 0.90^*$ (126K) |
| Polynormer | $40.75 \pm 0.46^*$ (665K) | $34.86 \pm 0.11^*$ (2.0M) | $41.16 \pm 0.93^*$ (6.2M) | $83.31 \pm 0.50^*$ (983K) | $86.79 \pm 0.28^*$ (1.8M) | $80.94 \pm 0.62^*$ (2.4M) |
| SSGNN | $43.10 \pm 1.36$ **(77K)** | $42.31 \pm 0.74$ **(34K)** | $43.22 \pm 1.05$ **(32K)** | $84.33 \pm 0.001$ **(157K)** | $88.66 \pm 0.17$ **(48.5K)** | $82.18 \pm 0.21$ **(121k)** |

## 4.3 GRAPH CLASSIFICATION AND REGRESSION

We evaluate SSGNN on three graph-level datasets of varying sizes: ZINC Dwivedi et al. (2023), a small dataset with 12,000 molecular graphs, and two larger OGB datasets, MolHIV and MolPCBA Hu et al. (2020), containing 41,000 and 437,000 graphs respectively. Nodes represent atoms, and edges denote chemical bonds. For fair comparison, we match Specformer's training settings. Results in Table 5 show that SSGNN outperforms SOTA models like GIN, PNA, SAN, GPS, and

Table 4: Average results $\pm$ std for node classification over 10 runs on heterophilic datasets. Accuracy is reported for roman-empire and amazon-ratings; ROC AUC is reported for the rest. The top **first**, **second**, and **third** results are highlighted for each dataset.

| | roman-empire | amazon-ratings | minesweeper | tolokers | questions |
|---|---|---|---|---|---|
| GCN | $73.69 \pm 0.74$ | $48.70 \pm 0.63$ | $89.75 \pm 0.52$ | $83.64 \pm 0.67$ | $76.09 \pm 1.27$ |
| GraphSAGE | $85.74 \pm 0.67$ | $53.63 \pm 0.39$ | $93.51 \pm 0.57$ | $82.43 \pm 0.44$ | $76.44 \pm 0.62$ |
| GAT-sep | $88.75 \pm 0.41$ | $52.70 \pm 0.62$ | $93.91 \pm 0.35$ | $83.78 \pm 0.43$ | $76.79 \pm 0.71$ |
| H2GCN | $60.11 \pm 0.52$ | $36.47 \pm 0.23$ | $89.71 \pm 0.31$ | $73.35 \pm 1.01$ | $63.59 \pm 1.46$ |
| GPRGNN | $64.85 \pm 0.27$ | $44.88 \pm 0.34$ | $86.24 \pm 0.61$ | $72.94 \pm 0.97$ | $55.48 \pm 0.91$ |
| FSGNN | $79.92 \pm 0.56$ | $52.74 \pm 0.83$ | $90.08 \pm 0.70$ | $82.76 \pm 0.61$ | $78.86 \pm 0.92$ |
| GloGNN | $59.63 \pm 0.69$ | $36.89 \pm 0.14$ | $51.08 \pm 1.23$ | $73.39 \pm 1.17$ | $65.74 \pm 1.19$ |
| GGCN | $74.46 \pm 0.54$ | $43.00 \pm 0.32$ | $87.54 \pm 1.22$ | $77.31 \pm 1.14$ | $71.10 \pm 1.57$ |
| OrderedGNN | $77.68 \pm 0.39$ | $47.29 \pm 0.65$ | $80.58 \pm 1.08$ | $75.60 \pm 1.36$ | $75.09 \pm 1.00$ |
| $G^2$-GNN | $82.16 \pm 0.78$ | $47.93 \pm 0.58$ | $91.83 \pm 0.56$ | $82.51 \pm 0.80$ | $74.82 \pm 0.92$ |
| DIR-GNN | $91.23 \pm 0.32$ | $47.89 \pm 0.39$ | $87.05 \pm 0.69$ | $81.19 \pm 1.05$ | $76.13 \pm 1.24$ |
| tGNN | $79.95 \pm 0.75$ | $48.21 \pm 0.53$ | $91.93 \pm 0.77$ | $70.84 \pm 1.75$ | $76.38 \pm 1.79$ |
| GraphGPS | $82.00 \pm 0.61$ | $53.10 \pm 0.42$ | $90.63 \pm 0.67$ | $83.71 \pm 0.48$ | $71.73 \pm 1.47$ |
| Exphormer | $89.03 \pm 0.37$ | $53.51 \pm 0.46$ | $90.74 \pm 0.53$ | $83.77 \pm 0.78$ | $73.94 \pm 1.06$ |
| NodeFormer | $64.49 \pm 0.73$ | $43.86 \pm 0.35$ | $86.71 \pm 0.88$ | $78.10 \pm 1.03$ | $74.27 \pm 1.46$ |
| DIFFormer | $79.10 \pm 0.32$ | $47.84 \pm 0.65$ | $90.89 \pm 0.58$ | $83.57 \pm 0.68$ | $72.15 \pm 1.31$ |
| Specformer | $OOM^*$ | $OOM^*$ | $93.95 \pm 0.39^*$ (8.5K) | $85.01 \pm 0.48^*$ (8.5K) | $OOM^*$ |
| Polynormer | $92.15 \pm 0.58^*$ (9.9M) | $54.36 \pm 0.32^*$ (9.1M) | $97.12 \pm 0.30^*$ (10.5M) | $84.51 \pm 0.88^*$ (7.9M) | $78.83 \pm 0.71^*$ (6.7M) |
| SSGNN | $83.90 \pm 1.21$ **(77K)** | $52.46 \pm 0.70$ **(690K)** | $94.38 \pm 0.54$ **(3K)** | $86.37 \pm 0.46$ **(3K)** | $79.16 \pm 0.16$ **(12.5K)** |

Table 5: Average results $\pm$ std for graph classification and regression. $\downarrow$ means lower the better, and $\uparrow$ means higher the better. We highlight the top **first**, **second**, and **third** results for each dataset.

| Model | ZINC($\downarrow$) | MolHIV($\uparrow$) | MolPCBA($\uparrow$) |
|---|---|---|---|
| GCN | $0.367 \pm 0.011$ | $0.7599 \pm 0.0119$ | $0.2424 \pm 0.0034$ |
| GIN | $0.526 \pm 0.051$ | $0.7707 \pm 0.0149$ | $0.2703 \pm 0.0023$ |
| CIN | $0.079 \pm 0.006$ | $0.8094 \pm 0.0057$ | - |
| GIN-AK+ | $0.080 \pm 0.001$ | $0.7961 \pm 0.0119$ | $0.2930 \pm 0.0044$ |
| GSN | $0.101 \pm 0.010$ | $0.7799 \pm 0.0100$ | - |
| DGN | $0.168 \pm 0.003$ | $0.7970 \pm 0.0097$ | $0.2885 \pm 0.0030$ |
| PNA | $0.188 \pm 0.004$ | $0.7905 \pm 0.0132$ | $0.2838 \pm 0.0035$ |
| SAN | $0.139 \pm 0.006$ | $0.7785 \pm 0.0025$ | $0.2765 \pm 0.0042$ |
| Graphormer | $0.122 \pm 0.006$ | $0.7640 \pm 0.0022$ | $0.2643 \pm 0.0017$ |
| GPS | $0.070 \pm 0.004$ | $0.7880 \pm 0.0101$ | $0.2907 \pm 0.0028$ |
| Specformer | $066 \pm 0.003$ (555K) | $0.7889 \pm 0.0124$ (227K) | $0.2972 \pm 0.0023$ (3.02M) |
| SSGNN | $0.0592 \pm 0.008$ **(401K)** | $0.8014 \pm 0.0193$ **(194K)** | $0.3012 \pm 0.0350$ **(2.64M)** |

Specformer, achieving new benchmarks of 0.0592 MAE on ZINC and 0.3012 AP on MolPCBA. On MolHIV, we surpass most models, ranking second only to CIN in AUC-ROC.

## 4.4 ABLATIONS

This section explores the effect of hyperparameters $b$ and $\epsilon$ on learned filters. The parameter $b$ controls the spread of our Relative Gaussian Amplifier (ReGA), influencing the distribution of scores across spectral encodings. A higher $b$ results in smoother filters with uniform score distribution, while a lower $b$ concentrates scores on a few encodings, causing sharp amplitude spikes. Fig. 3 shows this effect on the WikiCS dataset, where lower $b$ values (e.g., $b$=2) lead to sudden spikes, which smooth out as $b$ increases to 10.

We now examine how varying $\epsilon$ affects the learned filters. A higher $\epsilon$ captures more oscillatory patterns in spectral encodings, leading to stronger and more distinct components. However, this can cause abrupt score assignments in ReGA, amplifying certain spectral components and suppressing

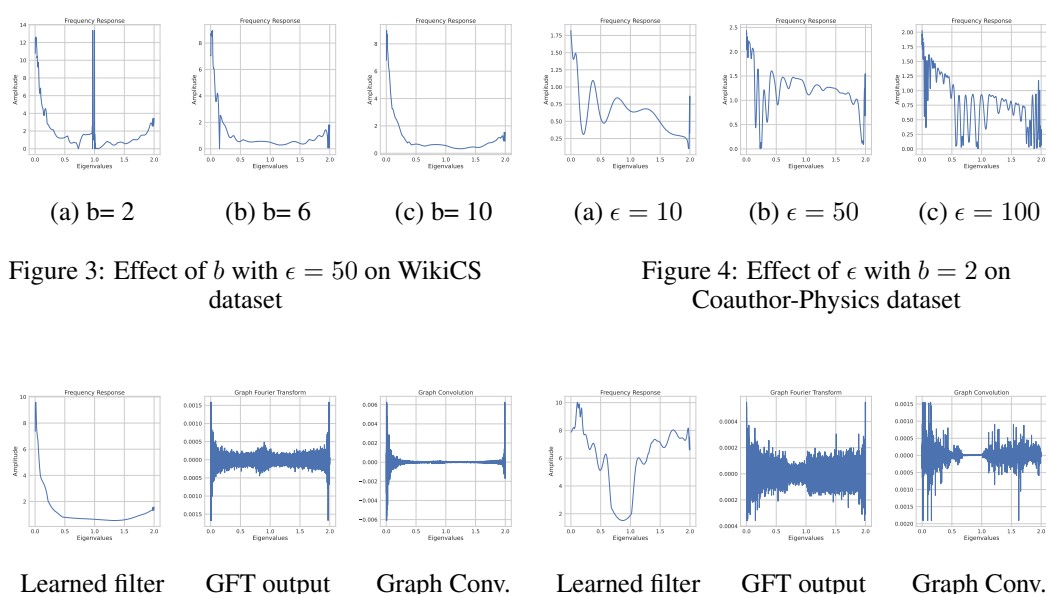

| (a) b= 2 | (b) b= 6 | (c) b= 10 | (a) $\epsilon = 10$ | (b) $\epsilon = 50$ | (c) $\epsilon = 100$ |

Figure 3: Effect of $b$ with $\epsilon = 50$ on WikiCS dataset

Figure 4: Effect of $\epsilon$ with $b = 2$ on Coauthor-Physics dataset

| Learned filter | GFT output | Graph Conv. | Learned filter | GFT output | Graph Conv. |

Figure 5: WikiCS dataset

Figure 6: Amazon-Ratings dataset

others, resulting in random spikes. Conversely, a lower $\epsilon$ produces smoother filters with fewer oscillations. Fig. 4 illustrates this effect on the Coauthor-Physics dataset, showing increased oscillations and amplitude spikes as $\epsilon$ rises from 10 to 100.

### 4.5 VISUALIZATIONS

In this section, we explore the filters learned by SSGNN and their impact through visualizing the Graph Fourier Transform (GFT) and the resulting convolved GFT output after applying our filters. These visualizations illustrate how our model effectively captures the underlying spectral properties of the graph, learning filters optimized for specific tasks. For the homophilic dataset WikiCS, our model learns an optimal low-pass filter (Fig. 5), allowing it to suppress high-frequency components and retain smooth node signals. In contrast, for the heterophilic dataset Amazon Ratings, SSGNN learns a band-reject filter (Fig. 6) which efficiently handles the more complex spectral characteristics of heterophilic graphs by filtering out irrelevant mid-frequency components. These results emphasize the critical role our learned filters play in the convolution process, adapting to the nature of the graph and enhancing performance across different types of data.

## 5 CONCLUSION

This work introduces SSGNN, a Simple Yet Effective Spectral Graph Neural network, powered by spectral encoding and a decoder supported by a novel parameter-free Relative Gaussian Amplifier (ReGA) to enhance adaptive filter learning. Synthetic data experiments show that SSGNN effectively captures complex spectral filters, while real-world results demonstrate that it achieves comparable or outperforms SOTA graph transformers and spectral GNNs. Notably, SSGNN achieves this with significant reductions in parameters, and runtime, highlighting its potential for advancing efficient spectral GNN research.

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

## A DATASET INFORMATION

Table 6: Statistics of Node Classification datasets used in our experiments.

| Dataset | Type | Nodes | Edges | Classes | Features |
|---|---|---|---|---|---|
| Cora | Homophily | $2,708$ | $5,429$ | 7 | $1,433$ |
| Citeseer | Homophily | $3,327$ | $4,732$ | 6 | $3,707$ |
| Computer | Homophily | $13,752$ | $245,861$ | 10 | 767 |
| Photo | Homophily | $7,650$ | $119,081$ | 8 | 745 |
| CS | Homophily | $18,333$ | $81,894$ | 15 | $6,805$ |
| Physics | Homophily | $34,493$ | $247,962$ | 5 | $8,415$ |
| WikiCS | Homophily | $11,701$ | $216,123$ | 10 | 300 |
| ogbn-arxiv | Homophily | $169,343$ | $1,166,243$ | 40 | 128 |
| Chameleon | Heterophily | 850 | $13,584$ | 5 | 2325 |
| Squirrel | Heterophily | $2,223$ | $65,718$ | 5 | 2089 |
| Penn94 | Heterophily | $41,554$ | $1,326,229$ | 2 | $4,814$ |
| roman-empire | Heterophily | $22,662$ | $32,927$ | 18 | 300 |
| amazon-ratings | Heterophily | $24,492$ | $93,050$ | 5 | 300 |
| minesweeper | Heterophily | $10,000$ | $39,402$ | 2 | 7 |
| tolokers | Heterophily | $11,758$ | $519,000$ | 2 | 10 |
| questions | Heterophily | $48,921$ | $153,540$ | 2 | 301 |

Table 7: Statistics of Graph Classification datasets used in our experiments.

| | Graphs | Avg. nodes | Avg. edges | Min nodes | Max nodes | Tasks | Metric |
|---|---|---|---|---|---|---|---|
| ZINC | 12,000 | 23.2 | 24.9 | 9 | 37 | Regression | MAE |
| MolHIV | 41,127 | 25.5 | 27.5 | 2 | 222 | Classification | AUROC |
| MolPCBA | 437,929 | 26.0 | 28.1 | 1 | 332 | Classification | AP |

## B  Our Compute

All experiments were conducted using two NVIDIA RTX 4090 GPUs, each with 24GB of memory. While the SSGNN model can be efficiently trained on a single GPU, both GPUs were utilized to train multiple configurations simultaneously by varying the hyperparameters $b$ and $\epsilon$. This parallel training approach significantly reduced the total training time. All models were implemented using the PyTorch deep learning framework.

## C  Stability Test against Purterbations.

Here we first provide the proof for the theoritical statement that we made in the main paper and later perform an ablation to justify the same (Table. 8).

**Theorem 1.** Let $\lambda \in \mathbb{R}^{N \times 1}$ be the corrected eigenvalue and $\tilde{\lambda} = \lambda + \xi$ represent a eigenvalue, where $\xi$ is a bounded pertubation such that $\|\xi\|_2 \leq \delta$. The Gaussian-based spectral filtering $f(\lambda) = \exp\left(-\frac{(\Lambda - \mu)^2}{2(\sigma c)^2}\right)$ remains stable under perturbations in the eigenvalue, such that:

$$\|f(\lambda) - f(\hat{\lambda})\|_2 \leq K\delta \tag{10}$$

where $K$ is a constant and $\delta$ is the perturbation bound.

*Proof.* Let $f(\lambda) = \exp\left(-\frac{(\lambda - \mu)^2}{2(\sigma c)^2}\right)$ represent the Gaussian-based spectral filtering function, where $\mu$ is the mean and $\sigma$ is the standard deviation of the eigenvalue vectors and $c$ is a scaling constant. Let $\xi$ represent a perturbation in the eigenvalues such that $\hat{\lambda} = \lambda + \xi$, with $\|\xi\|_2 \leq \delta$. We need to show that the filtered output $f(\lambda)$ remains stable under perturbations, that is, the difference between $f(\lambda)$ and $f(\hat{\lambda})$ is bounded.

Consider the difference between $f(\lambda)$ and $f(\hat{\lambda})$ as:

$$\left|f(\lambda_i) - f(\hat{\lambda}_i)\right| = \left|\exp\left(-\frac{(\lambda_i - \mu)^2}{2(\sigma c)^2}\right) - \exp\left(-\frac{(\hat{\lambda}_i - \mu)^2}{2(\sigma c)^2}\right)\right| \tag{11}$$

Since Gaussian is a smooth function over the interval $(-\infty, +\infty)$, we can use the Mean Value Theorem (MVT) to approximate the difference between the exponentials by their derivative

$$\left|f(\lambda_i) - f(\hat{\lambda}_i)\right| \leq \left|\frac{d}{d\lambda_i}\left(\exp\left(-\frac{(\lambda_i - \mu)^2}{2(\sigma c)^2}\right)\right)\right| \cdot \left|\lambda_i - \hat{\lambda}_i\right| \tag{12}$$

The derivative of the Gaussian function can be written as:

$$\frac{d}{d\lambda_i}\left(\exp\left(-\frac{(\lambda_i - \mu)^2}{2(\sigma c)^2}\right)\right) = -\frac{(\lambda_i - \mu)}{(\sigma c)^2}\exp\left(-\frac{(\lambda_i - \mu)^2}{2(\sigma c)^2}\right) \tag{13}$$

Hence, Eq. 12 can be rewritten as:

$$\left|f(\lambda_i) - f(\hat{\lambda}_i)\right| \leq \frac{|\lambda_i - \mu|}{(\sigma c)^2}\exp\left(-\frac{(\lambda_i - \mu)^2}{2(\sigma c)^2}\right) \cdot |\xi_i| \tag{14}$$

Since $|\lambda_i - \hat{\lambda}_i| = |\xi_i|$ and assuming $\|\xi\|_2 \leq \delta$, we can bound the right-hand side by a constant $K$ that depends on $\mu, \sigma$ and $c$ as

$$\left|f(\lambda_i) - f(\hat{\lambda}_i)\right| \leq K|\xi_i| \tag{15}$$

Since $\|\xi\|_2 \leq \delta$, the total deviation across all eigenvalue vectors is bounded by:

$$\|\mathbf{f}(\lambda) - \mathbf{f}(\hat{\lambda})\|_2 \leq K\|\xi\|_2 \leq K\delta \tag{16}$$

Here we compare the performance of SSGNN and Specformer when subjected to structural pertur­bations. We randomly remove 2%, 4%, 6%, 10% and 20% of edges from the original dataset and train both the models 10 times on across different splits / seeds. We then report the average accu­racy drop for both the models under different levels of perturbations. Table 8 shows that SSGNN remains significantly stable even under extreme perturbations across all datasets when compared to Specformer. Notably, SSGNN only underperforms on chameleon under a high perturbation.

Table 8: Perturbation analysis of SSGNN vs Specformer

|  | Drop Edges(%) | Cora | Citeseer | Chameleon | Squirrel |
|---|---|---|---|---|---|
| SSGNN | 2 | 0.51 | 0.66 | 0.54 | 0.46 |
|  | 4 | 0.63 | 0.56 | 0.19 | 0.43 |
|  | 6 | 0.66 | 0.58 | 0.32 | 0.32 |
|  | 10 | 1.38 | 0.62 | 0.51 | 0.21 |
|  | 20 | 1.51 | 0.92 | 0.88 | 0.52 |
| Specformer | 2 | 1.72 | 1.09 | 0.94 | 2.16 |
|  | 4 | 1.64 | 1.25 | 0.72 | 2.19 |
|  | 6 | 1.67 | 1.24 | 1.06 | 2.61 |
|  | 10 | 2.36 | 1.32 | 0.23 | 2.11 |
|  | 20 | 2.89 | 1.43 | 0.54 | 2.25 |

Table 9: Comparision of GFLOPs for Polynormer and our proposed SSGNN model. Bolded values indicate better metric (lower the better).

| heightModel | Roman-Empire | Amazon-Ratings | Minesweeper | Tolokers | Questions |
|---|---|---|---|---|---|
| Polynormer | 170.64 | 158.62 | 84.22 | 81.27 | 294.13 |
| SSGNN | **1.71** | **5.22** | **0.03** | **0.03** | **0.60** |
| **Model** | **Computer** | **Photo** | **CS** | **Physics** | **WikiCS** |
| Polynormer | 56.14 | 47.22 | 150.82 | 133.48 | 69.58 |
| SSGNN | **0.37** | **0.20** | **4.04** | **8.37** | **0.131** |

## D   POLYNORMER CONFIGURATIONS

This section outlines the various configuration details and the corresponding highest test accuracy achieved by Polynormer, as presented in Table 3. Polynormer was trained for 1000 epochs on each dataset with extensive gridsearch for optimal hyper-parameters, including hidden dimension (h), local epochs (LE), global epochs (GE), learning rate (LR), local layers (LL), global layers (GL), weight decay (WD), and the number of heads (H). The best-performing configuration for each dataset was selected and trained 10 times to obtain the average accuracy. Table 9 compares the GFLOPs of Polynormer and SSGNN.

*For results reported in Table 2 and Table 4 (of main paper) we use the default configuration provided by the authors of Polynormer.*

## E   BASELINE CONFIGURATIONS

We compare our model against various SOTA GNNs, spectral GNNs, and graph Transformers (GT). For homophilic datasets, we use a 60%-20%-20% testing split, as given in (He et al., 2021). For most heterophilic datasets, we adopt a 50%-25%-25% split, following Platonov et al. (2023). Homophilic methods are executed 10 times, while heterophilic methods run across 10 splits, with mean accuracy and standard deviation reported. We utilize Specformer results for homophilic datasets and Platonov et al. (2023) for heterophilic datasets, while Graph Transformers results are sourced from Polynormer. Both Specformer and Polynormer are trained from scratch on all datasets. We experiment with hidden dimensions $d$ ranging from 32 to 512 for both SSGNN and Specformer, while for Polynormer from 32 to 64. The number of heads is varied from 1 to 4 for SSGNN and Specformer, and from 2 to 8 for Polynormer. Learning rates are explored in the range of $10^2$ to $10^4$ for all models. For Polynormer, we assess configurations with 5-7 local layers and 2-3 global layers, with detailed configurations provided in the appendix. In the node classification task, we focus on large-scale graphs, specifically ogbn-arXiv and Penn94, employing truncated spectral decomposition. For Penn94, we utilize eigenvectors corresponding to the smallest 3000 (low-frequency) and largest 3000 (high-frequency) eigenvalues. For arXiv, we select the smallest 5000 eigenvalues (low-frequency). These are based on the experimental findings that low-pass filtering is effective

Table 10: Polynormer Configurations and Accuracies for Chameleon dataset

| h | LE | GE | LR | LL | GL | WD | H | Accuracy (%) |
|---|---|---|---|---|---|---|---|---|
| 32 | 100 | 900 | 0.001 | 5 | 2 | 0.0 | 2 | 34.02 |
| 32 | 100 | 900 | 0.01 | 5 | 2 | 0.0 | 2 | 31.95 |
| 32 | 100 | 900 | 0.0001 | 5 | 2 | 0.0 | 2 | 35.05 |
| 64 | 100 | 900 | 0.001 | 5 | 2 | 0.0 | 2 | 40.72 |
| 32 | 100 | 900 | 0.001 | 5 | 2 | 0.0 | 4 | 42.78 |
| 64 | 100 | 900 | 0.001 | 5 | 2 | 0.0 | 4 | 42.78 |
| 32 | 100 | 900 | 0.001 | 5 | 2 | 0.0 | 8 | 35.56 |
| 64 | 100 | 900 | 0.001 | 5 | 2 | 0.0 | 8 | 36.08 |
| 64 | 100 | 900 | 0.001 | 7 | 2 | 0.0 | 8 | 35.56 |
| 64 | 100 | 900 | 0.001 | 5 | 3 | 0.0 | 8 | 30.92 |

Table 11: Polynormer Configurations and Accuracies for Squirrel dataset.

| h | LE | GE | LR | LL | GL | WD | H | Accuracy (%) |
|---|---|---|---|---|---|---|---|---|
| 32 | 100 | 900 | 0.001 | 5 | 2 | 0.0 | 2 | 34.73 |
| 32 | 100 | 900 | 0.01 | 5 | 2 | 0.0 | 2 | 34.73 |
| 32 | 100 | 900 | 0.0001 | 5 | 2 | 0.0 | 2 | 34.73 |
| 64 | 100 | 900 | 0.001 | 5 | 2 | 0.0 | 2 | 33.62 |
| 32 | 100 | 900 | 0.001 | 5 | 2 | 0.0 | 4 | 34.82 |
| 64 | 100 | 900 | 0.001 | 5 | 2 | 0.0 | 4 | 35.73 |
| 32 | 100 | 900 | 0.001 | 5 | 2 | 0.0 | 8 | 35.73 |
| 64 | 100 | 900 | 0.001 | 5 | 2 | 0.0 | 8 | 34.73 |
| 64 | 100 | 900 | 0.001 | 7 | 2 | 0.0 | 8 | 34.12 |
| 64 | 100 | 900 | 0.001 | 5 | 3 | 0.0 | 8 | 34.77 |

Table 12: Polynormer Configurations and Accuracies for Cora dataset.

| h | LE | GE | LR | LL | GL | WD | H | Accuracy (%) |
|---|---|---|---|---|---|---|---|---|
| 32 | 100 | 900 | 0.001 | 5 | 2 | 0.0 | 2 | 86.86 |
| 32 | 100 | 900 | 0.01 | 5 | 2 | 0.0 | 2 | 85.71 |
| 32 | 100 | 900 | 0.0001 | 5 | 2 | 0.0 | 2 | 83.57 |
| 64 | 100 | 900 | 0.001 | 5 | 2 | 0.0 | 2 | 87.02 |
| 32 | 100 | 900 | 0.001 | 5 | 2 | 0.0 | 4 | 86.20 |
| 64 | 100 | 900 | 0.001 | 5 | 2 | 0.0 | 4 | 85.05 |
| 32 | 100 | 900 | 0.001 | 5 | 2 | 0.0 | 8 | 87.84 |
| 64 | 100 | 900 | 0.001 | 5 | 2 | 0.0 | 8 | 87.84 |
| 64 | 100 | 900 | 0.001 | 7 | 2 | 0.0 | 8 | 86.04 |
| 64 | 100 | 900 | 0.001 | 5 | 3 | 0.0 | 8 | 85.22 |

Table 13: Polynormer Configurations and Accuracies for Citeseer dataset.

| h | LE | GE | LR | LL | GL | WD | H | Accuracy (%) |
|---|---|---|---|---|---|---|---|---|
| 32 | 100 | 900 | 0.001 | 5 | 2 | 0.0 | 2 | 81.99 |
| 32 | 100 | 900 | 0.01 | 5 | 2 | 0.0 | 2 | 80.49 |
| 32 | 100 | 900 | 0.0001 | 5 | 2 | 0.0 | 2 | 80.76 |
| 64 | 100 | 900 | 0.001 | 5 | 2 | 0.0 | 2 | 80.49 |
| 32 | 100 | 900 | 0.001 | 5 | 2 | 0.0 | 4 | 77.76 |
| 64 | 100 | 900 | 0.001 | 5 | 2 | 0.0 | 4 | 79.40 |
| 32 | 100 | 900 | 0.001 | 5 | 2 | 0.0 | 8 | 79.94 |
| 64 | 100 | 900 | 0.001 | 5 | 2 | 0.0 | 8 | 81.17 |
| 64 | 100 | 900 | 0.001 | 7 | 2 | 0.0 | 8 | 80.76 |
| 64 | 100 | 900 | 0.001 | 5 | 3 | 0.0 | 8 | 79.53 |

Table 14: Polynormer Configurations and Accuracies for Penn94 dataset.

| h | LE | GE | LR | LL | GL | WD | H | Accuracy (%) |
|---|---|---|---|---|---|---|---|---|
| 32 | 100 | 900 | 0.001 | 5 | 2 | 0.0 | 2 | 82.94 |
| 32 | 100 | 900 | 0.01 | 5 | 2 | 0.0 | 2 | 83.45 |
| 32 | 100 | 900 | 0.0001 | 5 | 2 | 0.0 | 2 | 82.56 |
| 64 | 100 | 900 | 0.001 | 5 | 2 | 0.0 | 2 | 83.72 |
| 32 | 100 | 900 | 0.001 | 5 | 2 | 0.0 | 4 | 83.67 |
| 64 | 100 | 900 | 0.001 | 5 | 2 | 0.0 | 4 | OOM |
| 32 | 100 | 900 | 0.001 | 5 | 2 | 0.0 | 8 | OOM |
| 64 | 100 | 900 | 0.001 | 5 | 2 | 0.0 | 8 | OOM |
| 64 | 100 | 900 | 0.001 | 7 | 2 | 0.0 | 8 | OOM |
| 64 | 100 | 900 | 0.001 | 5 | 3 | 0.0 | 8 | OOM |

Table 15: Polynormer Configurations and Accuracies for Actor dataset.

| h | LE | GE | LR | LL | GL | WD | H | Accuracy (%) |
|---|---|---|---|---|---|---|---|---|
| 32 | 100 | 900 | 0.001 | 5 | 2 | 0.0 | 2 | 42.05 |
| 32 | 100 | 900 | 0.01 | 5 | 2 | 0.0 | 2 | 40.15 |
| 32 | 100 | 900 | 0.0001 | 5 | 2 | 0.0 | 2 | 36.09 |
| 64 | 100 | 900 | 0.001 | 5 | 2 | 0.0 | 2 | 41.11 |
| 32 | 100 | 900 | 0.001 | 5 | 2 | 0.0 | 4 | 37.42 |
| 64 | 100 | 900 | 0.001 | 5 | 2 | 0.0 | 4 | 39.89 |
| 32 | 100 | 900 | 0.001 | 5 | 2 | 0.0 | 8 | 38.88 |
| 64 | 100 | 900 | 0.001 | 5 | 2 | 0.0 | 8 | 41.29 |
| 64 | 100 | 900 | 0.001 | 7 | 2 | 0.0 | 8 | 39.01 |
| 64 | 100 | 900 | 0.001 | 5 | 3 | 0.0 | 8 | 42.49 |

Table 16: Polynormer Configurations and Accuracies for OGBN-Arxiv dataset.

| h | LE | GE | LR | LL | GL | WD | H | Accuracy (%) |
|---|---|---|---|---|---|---|---|---|
| 32 | 100 | 900 | 0.001 | 5 | 2 | 0.0 | 2 | 70.56 |
| 32 | 100 | 900 | 0.01 | 5 | 2 | 0.0 | 2 | 70.86 |
| 32 | 100 | 900 | 0.0001 | 5 | 2 | 0.0 | 2 | 58.29 |
| 64 | 100 | 900 | 0.001 | 5 | 2 | 0.0 | 2 | 72.08 |
| 32 | 100 | 900 | 0.001 | 5 | 2 | 0.0 | 4 | 72.03 |
| 64 | 100 | 900 | 0.001 | 5 | 2 | 0.0 | 4 | OOM |
| 32 | 100 | 900 | 0.001 | 5 | 2 | 0.0 | 8 | OOM |
| 64 | 100 | 900 | 0.001 | 5 | 2 | 0.0 | 8 | OOM |
| 64 | 100 | 900 | 0.001 | 7 | 2 | 0.0 | 8 | OOM |
| 64 | 100 | 900 | 0.001 | 5 | 3 | 0.0 | 8 | OOM |

for homophilic datasets, while band-rejection or high-pass filtering performs better in heterophilic scenarios.

## F  VISUAL INSIGHTS OF THE MULT-HEADED DECODER

In this section, we visualize how different heads of the decoder affect the performance of SSGNN.

For the Amazon-Computers dataset, employing a single head results in an accuracy of 90.5116 with convergence achieved at the 849th epoch. Increasing to two heads improves the accuracy to 90.775 and reduces the convergence time to the 764th epoch. Finally, our configuration with four heads achieves the best performance, with an accuracy of 91.28 and convergence at the 609th epoch.

Figures  19,  20, and  21 illustrate the filtering patterns learned for the 1-head, 2-head, and 4-head configurations, respectively. The visualizations reveal that with four heads, the filters exhibit substantial variation during the initial epochs (e.g., the 25th epoch). By the 250th epoch, the filters start to align, and by the 575th epoch, they converge to similar patterns. This progression highlights how multiple heads enable the model to learn diverse filtering patterns in the early stages, ultimately accelerating convergence and improving overall performance.

## G  ADDITIONAL EXPERIMENTAL INSIGHTS (ABLATIONS)

Based on the valueable feedback received from the reviewers, we have added some additionla experimnets that showcase interesting insights. In Tables 19-24 bold values indicate better results.

### G.1  EIGEN (SPECTRAL) DECOMPOSITION COST FOR SSGNN

Table 17 presents the pre-computation (eigen decomposition) time for the graphs. It is important to note that this decomposition is performed only once. Consequently, the overall complexity of SSGNN is better represented as the sum of its forward pass complexity and the decomposition cost *amortized* over the number of uses in training and inference, rather than a direct summation of the two.

Table 17: Decomposition Time

| Dataset | Time(s) |
|---|---|
| Cora | 1.0162 |
| Citeseer | 1.8773 |
| Photo | 19.1895 |
| Computer | 112.5633 |
| CS | 234.1317 |
| Physics | 1954.2803 |
| WikiCS | 63.9295 |
| Chameleon | 3.0252 |
| Squirrel | 5.6846 |
| roman-empire | 491.1278 |
| amazon-ratings | 713.6977 |
| minesweeper | 38.3126 |
| tolokers | 59.3125 |
| questions | 6173.5986 |

### G.2  TRAINING TIME COMPARISION OF SPECFORMER, POLYNORMER AND SSGNN

Table 18 compares the training time per epoch (in milliseconds) for Specformer, Polynormer and SSGNN. The results indicate that SGGNN is significantly faster than Polynormer across all datasets. However, on the chameleon and squirrel datasets Specformer is faster than SGGNN. This is because these datasets have very few nodes (850 and 2223, respectively) and in such cases, Specformer's set-to-set approach is more efficient than SGGNN's global context filtering approach. Nevertheless,

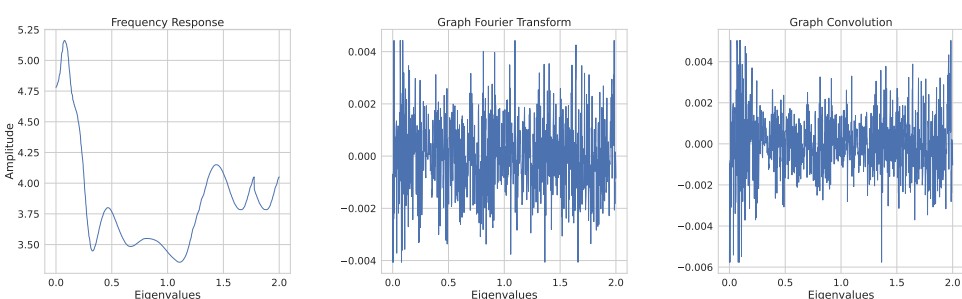

Figure 7: For Chameleon dataset, (left) the learned filter; (middle) GFT output; (right) Graph convolution output

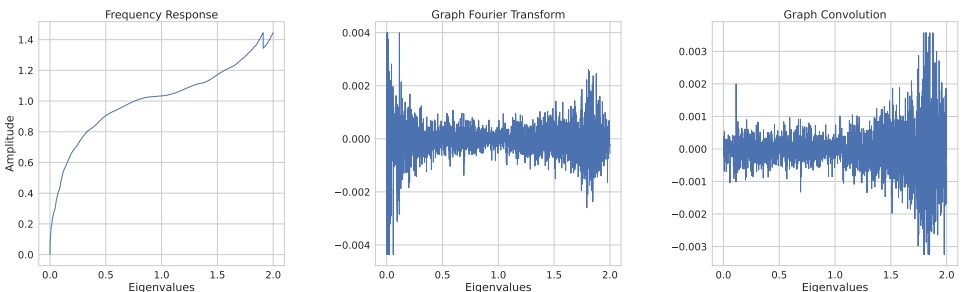

Figure 8: For Squirrel dataset, (left) the learned filter; (middle) GFT output; (right) Graph convolution output

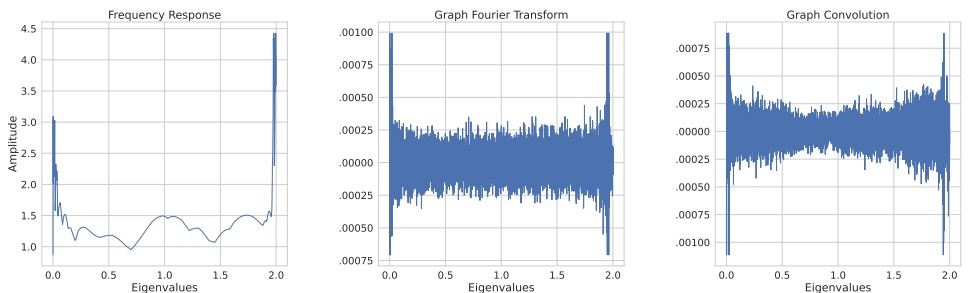

Figure 9: For Tolokers dataset, (left) the learned filter; (middle) GFT output; (right) Graph convolution output

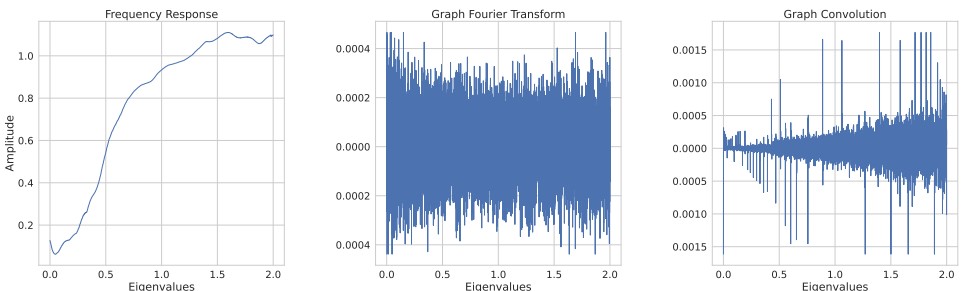

Figure 10: For Minesweeper dataset, (left) the learned filter; (middle) GFT output; (right) Graph convolution output

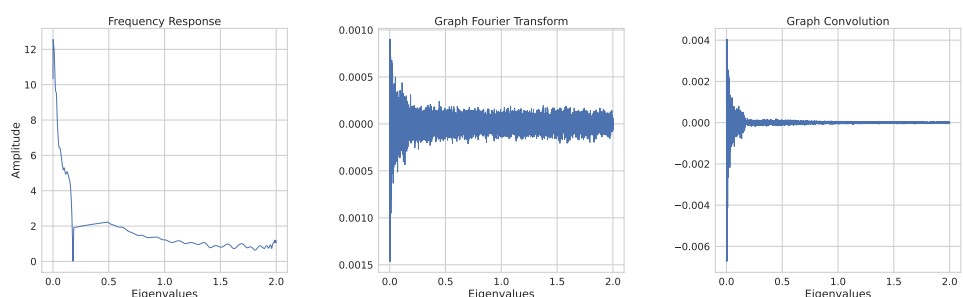

Figure 11: For CS dataset, (left) the learned filter; (middle) GFT output; (right) Graph convolution output

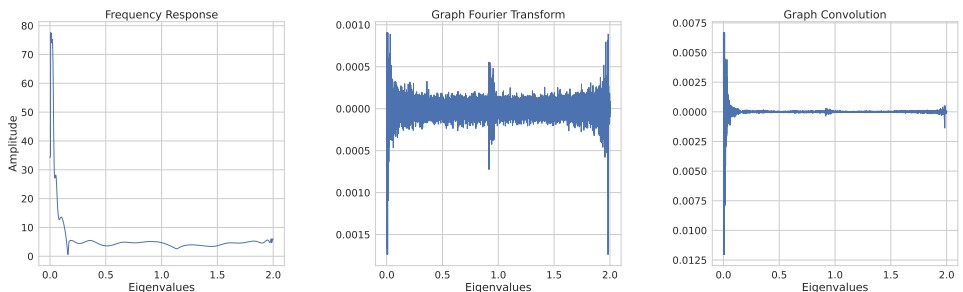

Figure 12: For Computer dataset, (left) the learned filter; (middle) GFT output; (right) Graph convolution output

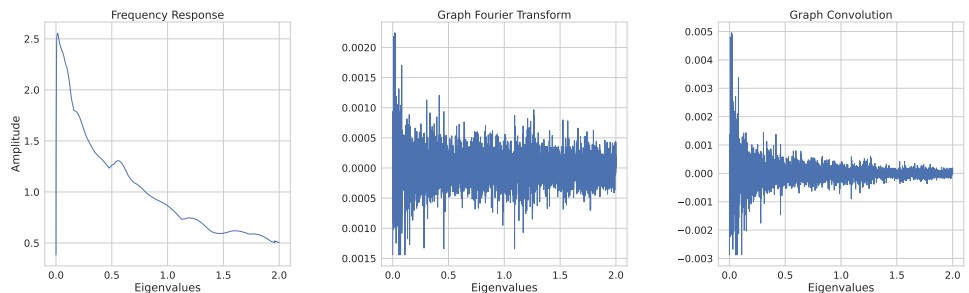

Figure 13: For Arxiv dataset, (left) the learned filter; (middle) GFT output; (right) Graph convolution output

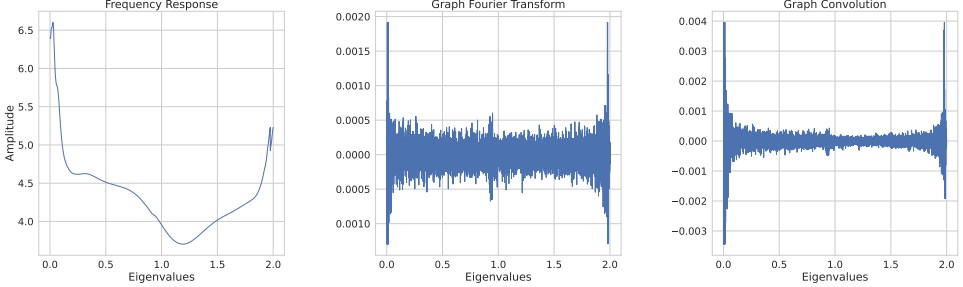

Figure 14: For Photo dataset, (left) the learned filter; (middle) GFT output; (right) Graph convolution output

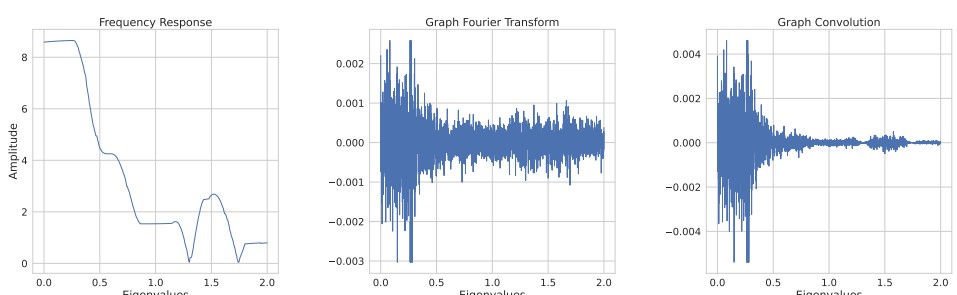

Figure 15: For Citeseer dataset, (left) the learned filter; (middle) GFT output; (right) Graph convolution output

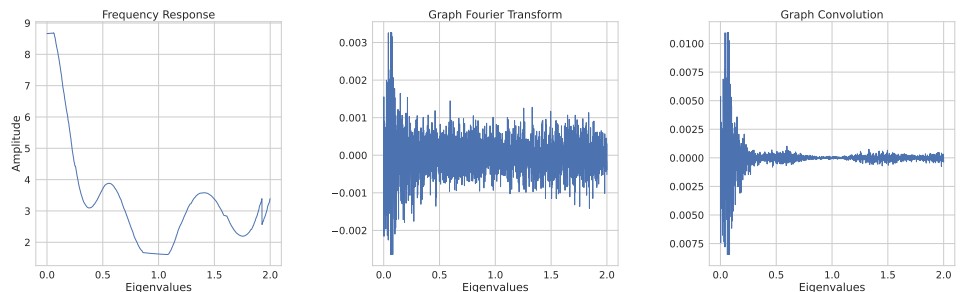

Figure 16: For Cora dataset, (left) the learned filter; (middle) GFT output; (right) Graph convolution output

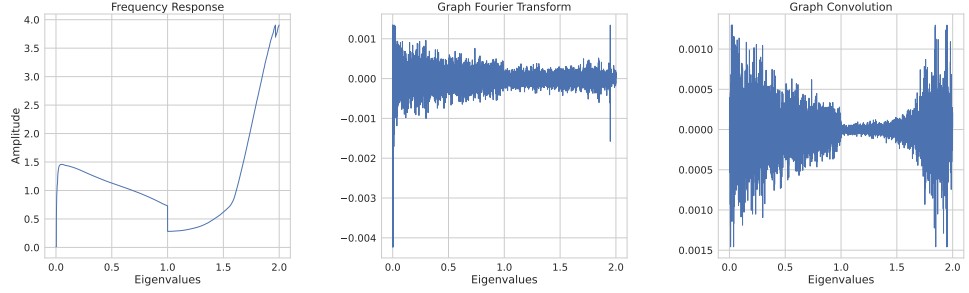

Figure 17: For Penn dataset, (left) the learned filter; (middle) GFT output; (right) Graph convolution output

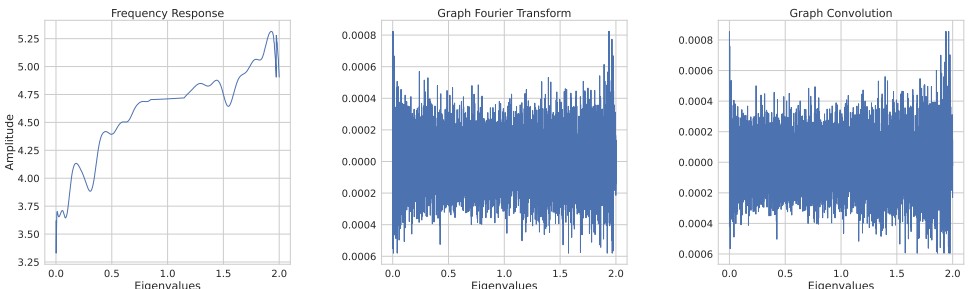

Figure 18: For Actor dataset, (left) the learned filter; (middle) GFT output; (right) Graph convolution output

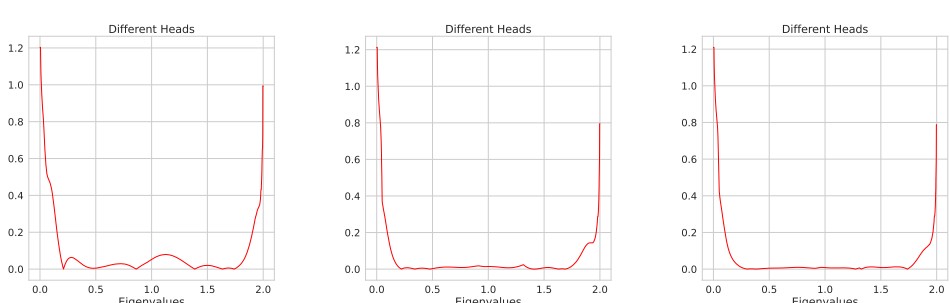

Figure 19: Filter learnt by decoder for Amazon Computer - 1 head configuration. (left) 25th epoch, (middle) 725th epoch, (right) 825th epoch

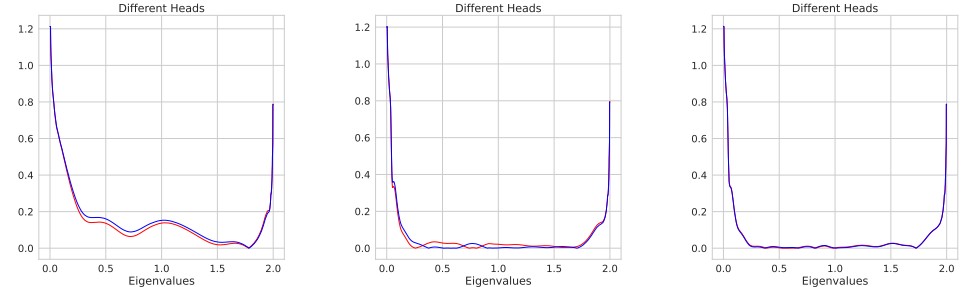

Figure 20: Filter learnt by decoder for Amazon Computer - 2 head configuration. (left) 25th epoch, (middle) 550th epoch, (right) 675th epoch

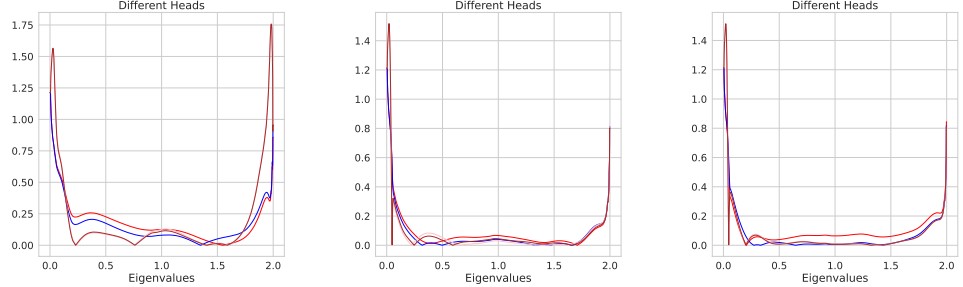

Figure 21: Filter learnt by decoder for Amazon Computer - 4 head configuration. (left) 25th epoch, (middle) 250th epoch, (right) 575th epoch

SGGNN outperforms Specformer in terms of speed on the remaining datasets and avoids the out-of-memory (OOM) issues that Specformer experiences.

Table 18: Training Time (ms) per epoch for Specformer, Polynormer, and SSGNN (ours). Bold values indicate lesser training time. We have also provided the number of parameters for each model on respective datasets.

| Type | Dataset | Specformer | Polynormer | SSGNN (Ours) |
|---|---|---|---|---|
| **Homophily** | Cora | 1.33 (54K) | 30.16 (1.8M) | **1.13** (48.5K) |
| | Citeseer | 1.35 (126K) | 33.20 (2.4M) | **0.98** (121K) |
| | CS | 67.94 (226K) | 23.48 (9.3M) | **8.33** (220K) |
| | Photo | 5.74 (32K) | 25.79 (7.8M) | **1.47** (26.5K) |
| | Computer | 21.31 (33K) | 19.13 (5.4M) | **7.35** (27.2K) |
| | Physics | *OOM* | 156.37 (4.0M) | **37.83** (135K) |
| | WikiCS | 12.95 (17.6K) | 69.39 (7.5M) | **2.86** (12.3K) |
| | ogbn-arxiv | 30.31 (500K) | 137.20 (393K) | **29.31** (36.5K) |
| **Heterophily** | Chameleon | **0.690** (82K) | 28.71 (665K) | 1.09 (77K) |
| | Squirrel | **0.697** (75K) | 27.63 (2.0M) | 1.12 (34K) |
| | Actor | 4.13 (37K) | 30.08 (6.2M) | **1.95** (32K) |
| | Penn94 | 7.78 (338K) | 41.44 (983K) | **6.90** (157K) |
| | roman-empire | *OOM* | 102.40 (9.9M) | **20.48** (77K) |
| | amazon-ratings | *OOM* | 127.11 (9.1M) | **91.80** (690K) |
| | minesweeper | 20.03 (8.5K) | 62.71 (10.5M) | **6.13** (3K) |
| | tolokers | 28.58 (8.5K) | 91.96 (7.9M) | **8.35** (3K) |
| | Questions | *OOM* | 360.22 (6.7M) | **8.35** (3K) |

## G.3 IS SPECFORMER BETTER WITHOUT A TRANSFORMER?

As mentioned in our abstract, we observed that specformer's performance improved without the transformer block on some datasets. This insight emerged as a key finding from our detailed investigation of Specformer. To evaluate the role of Transformers in spectral filtering, we performed ablation studies by removing the multi-head attention (MHA) and feedforward network (FFN) blocks, retaining only the EigenEncoder and decoder. Interestingly, this simplified architecture achieved results that were comparable to, or even better than, the Specformer, as shown in Table 19.

These findings highlight that Transformers contribute minimally to spectral domain performance, reinforcing the claim made in the abstract. Furthermore, the simplified architecture fits efficiently within the 24GB memory of an RTX 4090 GPU and delivers performance on par with state-of-the-art models. In contrast, the original Specformer is unable to fit on datasets such as Roman Empire, Amazon Ratings, and Questions.

Table 19: Results of Specformer, and Specformer-without-transformer. Specformer results are taken directly from our paper. We report avg results (%) ± std over 10 runs. ROC AUC is reported for Minesweeper, Tolokers, and Questions; accuracy is reported for the rest.

| Type | Dataset | Specformer | Specformer-w/o-transformer |
|---|---|---|---|
| **Homophily** | Computer | $87.23 \pm 0.52$ | $\mathbf{89.00} \pm 3.55$ |
| | Photo | $\mathbf{95.36} \pm 0.32$ | $95.32 \pm 0.38$ |
| | ogbn-arxiv | $71.98 \pm 0.33$ | $\mathbf{71.99} \pm 0.09$ |
| | Citeseer | $81.52 \pm 0.90$ | $\mathbf{81.84} \pm 0.95$ |
| | CS | $\mathbf{95.60} \pm 0.07$ | $95.31 \pm 0.33$ |
| | Cora | $\mathbf{88.50} \pm 0.98$ | $85.71 \pm 1.33$ |
| | WikiCS | $\mathbf{84.55} \pm 0.20$ | $84.50 \pm 0.46$ |
| | Physics | OOM | $\mathbf{98.02} \pm 0.38$ |
| **Heterophily** | Chameleon | $\mathbf{36.11} \pm 0.44$ | $35.98 \pm 2.04$ |
| | Squirrel | $37.66 \pm 0.42$ | $\mathbf{39.79} \pm 0.98$ |
| | Actor | $\mathbf{42.01} \pm 1.14$ | $41.69 \pm 1.30$ |
| | Penn94 | $84.28 \pm 0.32$ | $\mathbf{84.51} \pm 0.09$ |
| | roman-empire | OOM | $\mathbf{80.78} \pm 0.86$ |
| | amazon-ratings | OOM | $\mathbf{52.45} \pm 0.89$ |
| | Minesweeper | $\mathbf{93.95} \pm 0.39$ | $93.80 \pm 0.46$ |
| | Tolokers | $85.01 \pm 0.48$ | $\mathbf{85.18} \pm 0.67$ |
| | Questions | OOM | $\mathbf{76.69} \pm 1.24$ |

## G.4 EFFECT OF REGA ON SSGNN

To highlight the critical role of ReGA in SSGNN, we conduct an ablation study comparing the performance of SSGNN without ReGA to our proposed SSGNN, which incorporates ReGA. The results in Table 20 clearly demonstrate the impact of the ReGA module on performance across various datasets. Removing ReGA leads to a notable drop in performance, with improvements of 2.91% on Chameleon, 2.64% on Squirrel, 2.26% on Amazon Ratings, 2.07% on Actor, 1.69% on Cora, 1.39% on Penn94, 1.29% on Tolokers and 1.19% on Minesweeper when ReGA is included. These consistent gains across diverse datasets highlight that ReGA is not only integral but also a key contributor to SSGNN's SOTA performance.

## G.5 EFFECT OF RECENTERING (MEAN-SHIFT) ON REGA

To demonstrate the importance of recentering (Eq. 4), we conduct an ablation study where the mean shift was removed, and the decoder's output was directly passed to the scoring function. This resulted in a performance degradation, demonstrating that mean shift effectively captures global context and highlights the importance of interactions among eigenvalues. The results are shown in Table 21.

## G.6 EFFECT OF EIGEN-CORRECTION ON SPECFORMER AND SSGNN

We perform extensive ablations to evaluate the role of eigen-correction and determine whether the performance improvement of SSGNN primarily stems from ReGA or eigen-correction. Specifically, we trained Specformer with eigen-correction to ensure a fair comparison, as shown in Table 22. The results demonstrate that even with eigen-correction, Specformer is consistently outperformed by SSGNN across most datasets. Furthermore, we removed eigen-correction from SSGNN and observed its performance, as presented in Table 23. Notably, SSGNN without eigen-correction still outperformed Specformer in several datasets, confirming that SSGNN's ReGA module plays a critical role in its performance. Importantly, we observed that SSGNN gains significantly more from eigen-correction than Specformer, indicating its ability to leverage this enhancement. These results collectively demonstrate that while eigen-correction contributes to performance, SSGNN's primary advantage lies in its ReGA module, which drives its success independently of preprocessing tricks.

Table 20: Results of SSGNN-without-ReGA and SSGNN-with-ReGA (ours). We report avg results (%) ± std over 10 runs. ROC AUC is reported for Minesweeper, Tolokers, and Questions; accuracy is reported for the rest.

| Type | Dataset | SSGNN-w/o-ReGA | SSGNN-with-ReGA (Ours) |
|---|---|---|---|
| **Homophily** | Cora | $87.18 \pm 1.35$ | $\textbf{88.66} \pm 0.17$ |
| | Citeseer | $81.79 \pm 1.38$ | $\textbf{82.18} \pm 0.21$ |
| | Computer | $90.72 \pm 0.52$ | $\textbf{91.38} \pm 0.38$ |
| | Photo | $94.96 \pm 0.44$ | $\textbf{95.38} \pm 0.03$ |
| | CS | $95.21 \pm 0.41$ | $\textbf{96.30} \pm 0.08$ |
| | Physics | $97.47 \pm 0.40$ | $\textbf{98.33} \pm 0.15$ |
| | WikiCS | $84.54 \pm 0.33$ | $\textbf{85.16} \pm 0.41$ |
| | ogbn-arxiv | $\textbf{72.13} \pm 0.08$ | $72.10 \pm 0.04$ |
| **Heterophily** | Chameleon | $41.88 \pm 2.39$ | $\textbf{43.10} \pm 1.36$ |
| | Squirrel | $41.22 \pm 1.05$ | $\textbf{42.31} \pm 0.74$ |
| | Actor | $42.34 \pm 1.11$ | $\textbf{43.22} \pm 1.05$ |
| | Penn94 | $83.17 \pm 0.17$ | $\textbf{84.33} \pm 0.001$ |
| | roman-empire | $83.19 \pm 0.40$ | $\textbf{83.90} \pm 1.21$ |
| | amazon-ratings | $51.30 \pm 0.58$ | $\textbf{52.46} \pm 0.70$ |
| | Minesweeper | $93.27 \pm 0.38$ | $\textbf{94.38} \pm 0.54$ |
| | Tolokers | $85.02 \pm 0.89$ | $\textbf{86.37} \pm 0.46$ |
| | Questions | $78.15 \pm 1.10$ | $\textbf{79.16} \pm 0.16$ |

Table 21: Results of SSGNN and SSGNN-without-recentering. SSGNN results are taken directly from our paper. We report avg results (%) ± std over 10 runs. ROC AUC is reported for Minesweeper, Tolokers, and Questions; accuracy is reported for the rest.

| Type | Dataset | SSGNN-w/o-recentering | SSGNN |
|---|---|---|---|
| **Homophily** | Cora | $86.97 \pm 0.0143$ | $\textbf{88.66} \pm 0.17$ |
| | Citeseer | $81.20 \pm 0.013$ | $\textbf{82.18} \pm 0.21$ |
| | Computer | $90.18 \pm 0.006$ | $\textbf{91.38} \pm 0.38$ |
| | Photo | $94.63 \pm 0.003$ | $\textbf{95.38} \pm 0.03$ |
| | CS | $95.31 \pm 0.002$ | $\textbf{96.30} \pm 0.08$ |
| | Physics | $97.34 \pm 0.007$ | $\textbf{98.33} \pm 0.15$ |
| | WikiCS | $84.43 \pm 0.005$ | $\textbf{85.16} \pm 0.41$ |
| | ogbn-arxiv | $71.92 \pm 0.0008$ | $\textbf{72.10} \pm 0.04$ |
| **Heterophily** | Chameleon | $42.22 \pm 0.039$ | $\textbf{43.10} \pm 1.36$ |
| | Squirrel | $41.42 \pm 0.010$ | $\textbf{42.31} \pm 0.74$ |
| | Actor | $41.13 \pm 0.011$ | $\textbf{43.22} \pm 1.05$ |
| | Penn94 | $84.29 \pm 0.001$ | $\textbf{84.33} \pm 0.001$ |
| | roman-empire | $82.91 \pm 0.004$ | $\textbf{83.90} \pm 1.21$ |
| | amazon-ratings | $51.65 \pm 0.005$ | $\textbf{52.46} \pm 0.70$ |
| | Minesweeper | $93.63 \pm 0.003$ | $\textbf{94.38} \pm 0.54$ |
| | Tolokers | $84.84 \pm 0.006$ | $\textbf{86.37} \pm 0.46$ |
| | Questions | $78.09 \pm 0.012$ | $\textbf{79.16} \pm 0.16$ |

## G.7 COMPARISION WITH LIGHTWEIGHT SPECTRAL GNNS

We compare our proposed model SSGNN with lightweight spectral gnns like SGC (Wu et al. (2019)) and SSGC (Zhu & Koniusz (2021)). Table 24 demonstrates that SSGNN outperforms both SCG and SSGC across all datasets except chameleon. The results of SGC and SSGC were obtained after thorough hyper-parameter tuning. Table 25 shows the training time (in milliseconds) per epoch for SCG, SSGC and SSGNN. SGC and SSGC demonstrate faster training times due to their reliance on a simple two-layer MLP with precomputed aggregation. In contrast, SSGNN involves multiple components, including the encoder, decoder, ReGA, and graph convolution layer, which require

Table 22: Results of Specformer and SSGNN both with eigen-correction. SSGNN results are taken directly from our paper as we apply eigen-correction on all datasets. We report avg results (%) ± std over 10 runs. ROC AUC is reported for Minesweeper, Tolokers, and Questions; accuracy is reported for the rest.

| Type | Dataset | Specformer-with-eigen-corec. | SSGNN (Ours) |
|---|---|---|---|
| **Homophily** | Cora | $87.22 \pm 1.43$ | $\mathbf{88.66} \pm 0.17$ |
| | Citeseer | $81.04 \pm 1.18$ | $\mathbf{82.18} \pm 0.21$ |
| | Computer | $85.79 \pm 9.54$ | $\mathbf{91.38} \pm 0.38$ |
| | Photo | $\mathbf{95.69} \pm 0.235$ | $95.38 \pm 0.03$ |
| | CS | $95.40 \pm 0.26$ | $\mathbf{96.30} \pm 0.08$ |
| | Physics | OOM | $\mathbf{98.33} \pm 0.15$ |
| | WikiCS | $84.54 \pm 0.53$ | $\mathbf{85.16} \pm 0.41$ |
| | ogbn-arxiv | $72.07 \pm 0.073$ | $\mathbf{72.10} \pm 0.04$ |
| **Heterophily** | Chameleon | $36.12 \pm 3.83$ | $\mathbf{43.10} \pm 1.36$ |
| | Squirrel | $35.56 \pm 0.98$ | $\mathbf{42.31} \pm 0.74$ |
| | Actor | $41.44 \pm 1.12$ | $\mathbf{43.22} \pm 1.05$ |
| | Penn94 | $\mathbf{84.50} \pm 0.22$ | $84.33 \pm 0.001$ |
| | roman-empire | OOM | $\mathbf{83.90} \pm 1.21$ |
| | amazon-ratings | OOM | $\mathbf{52.46} \pm 0.70$ |
| | Minesweeper | $93.25 \pm 1.07$ | $\mathbf{94.38} \pm 0.54$ |
| | Tolokers | $84.85 \pm 0.84$ | $\mathbf{86.37} \pm 0.46$ |
| | Questions | OOM | $\mathbf{79.16} \pm 0.16$ |

Table 23: Results of Specformer and SSGNN both without eigen-correction. Specformer results are taken directly from our paper. We report avg results (%) ± std over 10 runs. ROC AUC is reported for Minesweeper, Tolokers, and Questions; accuracy is reported for the rest.

| Type | Dataset | Specformer-w/o-eigen-corec. | SSGNN-w/o-eigen-corec. |
|---|---|---|---|
| **Homophily** | Cora | $\mathbf{88.50} \pm 0.98$ | $88.45 \pm 1.48$ |
| | Citeseer | $81.52 \pm 0.90$ | $\mathbf{82.10} \pm 1.39$ |
| | Computer | $87.23 \pm 0.52$ | $\mathbf{90.98} \pm 0.34$ |
| | Photo | $\mathbf{95.36} \pm 0.32$ | $94.97 \pm 0.41$ |
| | CS | $\mathbf{95.60} \pm 0.07$ | $95.24 \pm 0.36$ |
| | Physics | OOM | $\mathbf{97.83} \pm 0.41$ |
| | WikiCS | $84.55 \pm 0.20$ | $\mathbf{85.04} \pm 0.21$ |
| | ogbn-arxiv | $71.98 \pm 0.33$ | $\mathbf{72.08} \pm 0.11$ |
| **Heterophily** | Chameleon | $36.11 \pm 0.44$ | $\mathbf{42.05} \pm 1.74$ |
| | Squirrel | $37.66 \pm 0.42$ | $\mathbf{42.15} \pm 1.15$ |
| | Actor | $42.01 \pm 1.14$ | $\mathbf{42.65} \pm 1.16$ |
| | Penn94 | $84.28 \pm 0.32$ | $\mathbf{84.35} \pm 0.12$ |
| | roman-empire | OOM | $\mathbf{83.43} \pm 0.56$ |
| | amazon-ratings | OOM | $\mathbf{51.87} \pm 0.53$ |
| | Minesweeper | $\mathbf{93.95} \pm 0.39$ | $93.78 \pm 0.87$ |
| | Tolokers | $85.01 \pm 0.48$ | $\mathbf{85.43} \pm 0.95$ |
| | Questions | OOM | $\mathbf{78.47} \pm 1.35$ |

more computation time compared to SGC and SSGC's minimal architecture. However, our findings highlight a critical distinction: while SGC and SSGC perform well on homophilic graphs, their precomputation method, rooted in homophilic aggregation, hampers their effectiveness in heterophilic settings. SSGNN, on the other hand, effectively models both homophilic and heterophilic patterns, achieving a balanced tradeoff between accuracy, training speed, parameter efficiency, and computational demands.

Table 24: Results of SGC, SSGC, and SSGNN (ours)

| Type | Dataset | SGC | SSGC | SSGNN (Ours) |
|---|---|---|---|---|
| **Homophily** | Cora | $87.65 \pm 0.021$ | $88.34 \pm 0.057$ | $\textbf{88.66} \pm 0.17$ |
| | Citeseer | $81.51 \pm 0.015$ | $81.99 \pm 0.034$ | $\textbf{82.18} \pm 0.21$ |
| | Computer | $86.93 \pm 0.011$ | $87.71 \pm 0.007$ | $\textbf{91.38} \pm 0.38$ |
| | Photo | $92.44 \pm 0.006$ | $93.11 \pm 0.005$ | $\textbf{95.38} \pm 0.03$ |
| | CS | $93.23 \pm 0.041$ | $93.77 \pm 0.052$ | $\textbf{96.30} \pm 0.08$ |
| | Physics | $97.64 \pm 0.016$ | $98.11 \pm 0.022$ | $\textbf{98.33} \pm 0.15$ |
| | WikiCS | $74.19 \pm 0.023$ | $75.02 \pm 0.018$ | $\textbf{85.16} \pm 0.41$ |
| | ogbn-arxiv | $60.69 \pm 0.027$ | $61.04 \pm 0.038$ | $\textbf{72.10} \pm 0.04$ |
| **Heterophily** | Chameleon | $44.84 \pm 0.032$ | $\textbf{45.34} \pm 0.029$ | $43.10 \pm 1.36$ |
| | Squirrel | $40.23 \pm 0.031$ | $40.42 \pm 0.030$ | $\textbf{42.31} \pm 0.74$ |
| | Actor | $26.43 \pm 0.010$ | $27.63 \pm 0.008$ | $\textbf{43.22} \pm 1.05$ |
| | Penn94 | $80.01 \pm 0.031$ | $80.43 \pm 0.043$ | $\textbf{84.33} \pm 0.001$ |
| | roman-empire | $52.20 \pm 0.037$ | $53.08 \pm 0.029$ | $\textbf{83.90} \pm 1.21$ |
| | amazon-ratings | $43.83 \pm 0.046$ | $44.73 \pm 0.032$ | $\textbf{52.46} \pm 0.70$ |
| | minesweeper | $83.41 \pm 0.042$ | $84.22 \pm 0.021$ | $\textbf{94.38} \pm 0.54$ |
| | tolokers | $78.84 \pm 0.019$ | $79.35 \pm 0.027$ | $\textbf{86.37} \pm 0.46$ |

Table 25: Training Time (ms) per epoch of SGC, SSGC, and SSGNN (ours). Bold values indicate faster training time. We have also provided number of parameters for each model across all datasets.

| Type | Dataset | SGC | SSGC | SSGNN (Ours) |
|---|---|---|---|---|
| **Homophily** | Cora | 0.531 (11.5K) | **0.526** (11.5K) | 1.13 (48.5K) |
| | Citeseer | **0.495** (29.8K) | **0.495** (29.8K) | 0.98 (121K) |
| | CS | 0.602 (54.5K) | **0.598** (54.5K) | 8.33 (220K) |
| | Photo | 0.563 (6K) | **0.560** (6K) | 1.47 (26.5K) |
| | Computer | 0.584 (6.2K) | **0.581** (6.2K) | 7.35 (27.2K) |
| | Physics | 0.562 (67.3K) | **0.558** (67.3K) | 37.83 (135K) |
| | WikiCS | **0.559** (2.4K) | 0.567 (2.4K) | 2.86 (12.3K) |
| | ogbn-arxiv | 0.742 (1.3K) | **0.647** (1.3K) | 29.31 (36.5K) |
| **Heterophily** | Chameleon | 0.690 (18.6k) | **0.659** (18.6K) | 1.09 (77K) |
| | Squirrel | **0.697** (16.7K) | 0.761 (16.7K) | 1.12 (34K) |
| | Actor | 0.695 (7.5K) | **0.663** (7.5K) | 1.95 (32K) |
| | Penn94 | **0.635** (38.5K) | 0.7394 (38.5K) | 6.90 (157K) |
| | roman-empire | **0.656** (2.5K) | 0.665 (2.5K) | 20.48 (77K) |
| | amazon-ratings | **0.677** (2.4K) | 0.678 (2.4K) | 91.80 (690K) |
| | minesweeper | **0.905** (82) | 0.919 (82) | 6.13 (3K) |
| | tolokers | **0.655** (106) | 0.658 (106) | 8.35 (3K) |

