# OpenReview forum: "SSGNN: Simple Yet Effective Spectral Graph Neural Network"
_ICLR.cc/2025/Conference — Submitted to ICLR 2025_

### Official Review · Reviewer_kCNh · 2024-10-29

**Soundness:** 3
**Presentation:** 2
**Contribution:** 3
**Rating:** 5
**Confidence:** 4

**Summary:**

This paper proposes SSGNN, a simple and effective GNN model which can achieve good performance with much reduced parameters compared to transformers. With basic spectral encoder-decoder structure, it further incorporates a REGA module to strengthen the representational capabilities and the robustness against spectral perturbation. Experiments demonstrate its effectiveness and superiority on model parameters.

**Strengths:**

1-	The method is simple and effective on multiple graph downstream tasks.

2-	The experiments are comprehensive and solid.

3-	The theoretical analysis of ReGA is interesting and novel.

**Weaknesses:**

1-	The time and space cost in pre-computation and training stage of SSGNN seems to be a bottleneck for large graphs. Though top-k techniques can be applied, it’ll lose spectral frequencies which is essential for down-streaming tasks. When comparing parameter amounts and GFLOPS, it’ll be fair to show the training and pre-computation cost for baselines, transformers and other GNNs.

2-	It needs experiments to demonstrate the contribution of ReGa, which is the most novel part in SSGNN. Will the removal of it greatly influence the performance?

3-	Presentation can be improved. “. .” appears in line 373 and line 398 appears incomplete sentences “* means” what?

**Questions:**

In the abstract, “Our analysis indicates that applying Transformers to these filters provides minimal advantage in the spectral domain.” While such analysis seems to be missing in the content. Could you elaborate it more?

---

> ### Author Response · Authors · 2024-11-21
> **Clarification regarding pre-computation time(s) for SSGNN and training time (ms) comparison**
>
> To address concerns regarding sparse decomposition, we leverage insights from the learned filters: band-rejection and high-pass filters are crucial for heterophilic datasets, while low-pass filters work well for homophilic datasets. Accordingly, we use eigenvectors with the smallest 3000 (low-frequency) and largest 3000 (high-frequency) eigenvalues for Penn94 (large-scale heterophilic) and the smallest 5000 eigenvalues (low-frequency) for ogbn-arXiv (large-scale homophilic). This targeted approach ensures meaningful spectral decomposition, avoiding performance loss while maintaining computational efficiency. Experimental results in Tables 2 and 3 of the main paper demonstrate that under these settings, SSGNN outperforms other models on ogbn-arXiv (large-scale homophilic graph) and Penn94 (large-scale heterophilic graph), demonstrating the effectiveness of this strategic spectral decomposition.
>
> Table 1 presents the pre-computation (eigen decomposition) time for the graphs. It is important to note that this **decomposition is performed only once**. Consequently, the **overall complexity of SSGNN** is better represented as the sum of its forward pass complexity and the decomposition cost **amortized** over the number of uses in training and inference, rather than a direct summation of the two.
>
> **Table-1**: Precomputation (decomposition) time for different datasets.
> | Dataset         | Time(s)     |
> |-----------------|-------------|
> | Cora            | 1.0162      |
> | Citeseer        | 1.8773      |
> | Photo           | 19.1895     |
> | Computer        | 112.5633    |
> | CS              | 234.1317    |
> | Physics         | 1954.2803   |
> | WikiCS          | 63.9295     |
> | Chameleon       | 3.0252      |
> | Squirrel        | 5.6846      |
> | roman-empire    | 491.1278    |
> | amazon-ratings  | 713.6977    |
> | minesweeper     | 38.3126     |
> | tolokers        | 59.3125     |
> | questions       | 6173.5986   |
>
> Table 2 summarizes the training time per epoch (ms) for Specformer, Polynormer, and SSGNN across homophilic and heterophilic graphs. **SSGNN** consistently demonstrates **faster** training times compared to Specformer and Polynormer, even on **large-scale graphs like ogbn-arxiv and Penn94**. However, on smaller datasets such as Chameleon and Squirrel, Specformer achieves slightly faster training times, likely due to the smaller node count.
>
> $\textbf{Table-2:}$ Training time per epoch (in ms) of Specformer, Polynormer and SSGNN.
> | Type         | Dataset         | Specformer       | Polynormer       | SSGNN (Ours)     |
> |--------------|-----------------|------------------|------------------|------------------|
> | **Homophily** | Cora            | 1.33             | 30.16            | **1.13**             |
> |              | Citeseer        | 1.35             | 33.20            | **0.98**             |
> |              | CS              | 67.94            | 23.48            | **8.33**             |
> |              | Photo           | 5.74             | 25.79            | **1.47**        |
> |              | Computer        | 21.31            | 19.13            | **7.35**             |
> |              | Physics         | OOM              | 156.37           | **37.83**            |
> |              | WikiCS          | 12.95            | 69.39            | **2.86**            |
> |              | ogbn-arxiv (large-scale)      | 30.31            | 137.20           | **29.31**            |
> |              |                 |                  |                  |                  |
> | **Heterophily** | Chameleon     | **0.69**            | 28.71            | 1.09             |
> |              | Squirrel        | **0.70**            | 27.63            | 1.12             |
> |              | Actor           | 4.13             | 30.08            | **1.95**             |
> |              | Penn94 (large-scale)          | 7.78             | 41.44            | **6.90**            |
> |              | roman-empire    | OOM              | 102.40           | **20.48**            |
> |              | amazon-ratings  | OOM              | 127.11           | **91.80**            |
> |              | minesweeper     | 20.03            | 62.71            | **6.13**             |
> |              | tolokers        | 28.58            | 91.96            | **8.35**             |
> |              | Questions       | OOM              | 360.22           | **128.47**             |
>
> This addresses the reviewer’s concerns regarding the inclusion of training costs for baseline models and the precomputation time (eigen decomposition) required for SSGNN.

---

> ### Author Response · Authors · 2024-11-21
> **Impact of ReGA**
>
> The results in the table below clearly illustrate the significant impact of the ReGA module on performance across various datasets.
>
> | Type         | Dataset         | SSGNN-without-ReGA         | SSGNN-with-ReGA (Ours)    |
> |--------------|-----------------|----------------------------|---------------------------|
> | **Homophily** | Cora            | 87.18 $\pm$ 1.35              | **88.66 $\pm$ 0.17**          |
> |              | Citeseer        | 81.79 $\pm$ 1.38              | **82.18 $\pm$ 0.21**          |
> |              | Computer        | 90.72 $\pm$ 0.52              | **91.38 $\pm$ 0.38**          |
> |              | Photo           | 94.96 $\pm$ 0.44              | **95.38 $\pm$ 0.03**          |
> |              | CS              | 95.21 $\pm$ 0.41              | **96.30 $\pm$ 0.08**          |
> |              | Physics         | 97.47 $\pm$ 0.40              | **98.33 $\pm$ 0.15**          |
> |              | WikiCS          | 84.54 $\pm$ 0.33              | **85.16 $\pm$ 0.41**          |
> |              | ogbn-arxiv (large-scale)      | **72.13 $\pm$ 0.08**          | 72.10 $\pm$ 0.04              |
> |              |                 |                           |                           |
> | **Heterophily** | Chameleon     | 41.88 $\pm$ 2.39              | **43.10 $\pm$ 1.36**          |
> |              | Squirrel        | 41.22 $\pm$ 1.05              | **42.31 $\pm$ 0.74**          |
> |              | Actor          | 42.34 $\pm$ 1.11              | **43.22 $\pm$ 1.05**          |
> |              | Penn94 (large-scale)          | 83.17 $\pm$ 0.17              | **84.33 $\pm$ 0.001**         |
> |              | roman-empire    | 83.19 $\pm$ 0.40              | **83.90 $\pm$ 1.21**          |
> |              | amazon-ratings  | 51.30 $\pm$ 0.58              | **52.46 $\pm$ 0.70**          |
> |              | Minesweeper     | 93.27 $\pm$ 0.38              | **94.38 $\pm$ 0.54**          |
> |              | Tolokers        | 85.02 $\pm$ 0.89              | **86.37 $\pm$ 0.46**          |
> |              | Questions       | 78.15 $\pm$ 1.10              | **79.16 $\pm$ 0.16**          |
>
> **Removing ReGA** results in a noticeable **decline** in accuracy, with performance **improvements of 2.91% on Chameleon, 2.64% on Squirrel, 2.26% on Amazon Ratings, 2.07% on Actor, 1.69% on Cora, 1.39% on Penn94, 1.29% on Tolokers, and 1.19% on Minesweeper** when ReGA is **incorporated**. These consistent improvements across various datasets highlight ReGA’s pivotal role as a **key contributor** to SSGNN’s performance, directly addressing the reviewer’s concern regarding the impact of the ReGA module.
>
> $\textbf{Answer-3}$: Thank you for pointing this out. We will definitely try to improve the manuscript's presentation. Just to clarify, “*” indicates models that were trained from scratch.

---

> ### Author Response · Authors · 2024-11-21
> **Transformers provide minimial performance advantage at the spectrum level**
>
> This insight emerged as a key finding from our detailed investigation of Specformer. To evaluate the role of Transformers in spectral filtering, we performed ablation studies by **removing** the **multi-head attention (MHA)** and **feedforward network (FFN)** blocks, retaining only the EigenEncoder and decoder. Interestingly, this simplified architecture achieved **results** that were **comparable to, or even better than, the Specformer**, as shown in the table below.
>
> | Type         | Dataset         | Specformer  $~~~~~$           | Specformer-without-transformer  |
> |--------------|-----------------|--------------------------|----------------------------------|
> | **Homophily**|   Cora            | **88.50 $\pm$ 0.98**         | 85.71 $\pm$ 1.33        |
> |              | Citeseer        | 81.52 $\pm$ 0.90            | **81.84 $\pm$ 0.95**                |
> |              | Computer        | 87.23 $\pm$ 0.52            | **89.00 $\pm$ 3.55**                |
> |              | Photo           | **95.36 $\pm$ 0.32**         | 95.32 $\pm$ 0.38                    |
> |              | CS              | **95.60 $\pm$ 0.07**         | 95.31 $\pm$ 0.33                    |
> |              | Physics         | OOM                     | **98.02 $\pm$ 0.38**                |
> |              | WikiCS          | **84.55 $\pm$ 0.20**         | 84.50 $\pm$ 0.46                    |
> |              | ogbn-arxiv      | 71.98 $\pm$ 0.33            | **71.99 $\pm$ 0.09**                |
> | **Heterophily** | Chameleon     | **36.11 $\pm$ 0.44**         | 35.98 $\pm$ 2.04                    |
> |              | Squirrel        | 37.66 $\pm$ 0.42            | **39.79 $\pm$ 0.98**                |
> |              | Actor           | **42.01 $\pm$ 1.14**         | 41.69 $\pm$ 1.30                    |
> |              | Penn94          | 84.28 $\pm$ 0.32            | **84.51 $\pm$ 0.09**                |
> |              | roman-empire    | OOM                     | **80.78 $\pm$ 0.86**                |
> |              | amazon-ratings  | OOM                     | **52.45 $\pm$ 0.89**                |
> |              | Minesweeper     | **93.95 $\pm$ 0.39**         | 93.80 $\pm$ 0.46                    |
> |              | Tolokers        | 85.01 $\pm$ 0.48            | **85.18 $\pm$ 0.67**                |
> |              | Questions       | OOM                     | **76.69 $\pm$ 1.24**                |
>
> These findings highlight that Transformers contribute minimally to spectral domain performance, reinforcing the claim made in the abstract. Furthermore, the simplified architecture fits efficiently within the 24GB memory of an RTX 4090 GPU and delivers performance on par with SOTA models. In contrast, the original Specformer is unable to fit on datasets such as Roman Empire, Amazon Ratings, and Questions.

---

### Official Review · Reviewer_SN5E · 2024-11-03

**Soundness:** 2
**Presentation:** 2
**Contribution:** 2
**Rating:** 5
**Confidence:** 4

**Summary:**

This paper proposes a Simple Yet Effective Spectral Graph Neural Network (SSGNN), which simplifies the set-to-set graph filter, e.g., Specformer, without performance degeneration. The key component is a parameter-free Relative Gaussian Amplifier (ReGA), which not only improves model performance but also maintains the robustness against graph perturbations. Extensive experiments on both node-level and graph-level datasets demonstrate the superiority of SSGNN in terms of effectiveness and efficiency over baselines.

**Strengths:**

1. The design of the ReGA component in SSGNN has some novelty. It not only avoids negative eigenvalues but also improves the robustness of SSGNN.

2. This paper conducts extensive experiments and SSGNN also shows competitive performance. For example, in the ZINC dataset, SSGNN has a RMSE value of 0.0592.

**Weaknesses:**

1. This paper claims that SSGNN uses a **simplified set-to-set approach** to capture key spectral features. However, there is no interaction between different eigenvalues in SSGNN. Specifically, the matrix $Z_{eig} \in \mathbb{R}^{N \times (d+1)}$ indicates the $(d+1)$ dimensional representation of each eigenvalue. The transformation $W_{eig}$ is applied on the channels of a single eigenvalue, i.e., $Z_{eig}W_{eig}$. Therefore, SSGNN is not a set-to-set approach.

2. SSGNN involves a lot of tricks in the training process, such as eigen-correction. However, both Specformer and Polynormer do not use eigen-correction for data preprocessing. In this case, it is necessary to make a comprehensive ablation study to validate the roles of each trick. We need to verify whether the performance improvement mainly comes from ReGA rather than eigen-correction.

3. It would be better if the authors could provide a comparison of the time and space overhead between different methods.

4. How many parameters does SSGNN have in Table 5? Generally, we will control the number of parameters around 50K in the ZINC dataset.

**Questions:**

See weaknesses.

---

> ### Author Response · Authors · 2024-11-25
> **Clarification regarding set-to-set filtering**
>
> We acknowledge the reviewer’s observation regarding the absence of explicit set-to-set modelling in SSGNN. Unlike Specformer, which employs transformers to model relative interactions among eigenvalues, SSGNN achieves **global context filtering through the mean shift** applied to the decoder’s output. This mean shift computes a global mean and standard deviation across all N nodes, **implicitly enabling interaction** among eigenvalues.
>
> To further substantiate the importance of this mechanism, we conducted an ablation study where the **mean shift was removed**, and the decoder’s output was directly passed to the scoring function. This resulted in a clear performance **degradation** as shown in the Table below, demonstrating that mean shift effectively **captures global context** and highlights the importance of **interactions among eigenvalues**.
>
> | **Type**       | **Dataset**      | **SSGNN-without-recentering** | **SSGNN**              |
> |-----------------|------------------|------------------------------|------------------------|
> | **Homophily**   | Cora            | 86.97 $\pm$ 0.0143           | **88.66 $\pm$ 0.17**   |
> |                 | Citeseer        | 81.20 $\pm$ 0.013            | **82.18 $\pm$ 0.21**   |
> |                 | Computer        | 90.18 $\pm$ 0.006            | **91.38 $\pm$ 0.38**   |
> |                 | Photo           | 94.63 $\pm$ 0.003            | **95.38 $\pm$ 0.03**   |
> |                 | CS              | 95.31 $\pm$ 0.002            | **96.30 $\pm$ 0.08**   |
> |                 | Physics         | 97.34 $\pm$ 0.007            | **98.33 $\pm$ 0.15**   |
> |                 | WikiCS          | 84.43 $\pm$ 0.005            | **85.16 $\pm$ 0.41**   |
> |                 | ogbn-arxiv      | 71.92 $\pm$ 0.0008           | **72.10 $\pm$ 0.04**   |
> | **Heterophily** | Chameleon       | 42.22 $\pm$ 0.039            | **43.10 $\pm$ 1.36**   |
> |                 | Squirrel        | 41.42 $\pm$ 0.010            | **42.31 $\pm$ 0.74**   |
> |                 | Actor           | 41.13 $\pm$ 0.011            | **43.22 $\pm$ 1.05**   |
> |                 | Penn94          | 84.29 $\pm$ 0.001            | **84.33 $\pm$ 0.001**  |
> |                 | roman-empire    | 82.91 $\pm$ 0.004            | **83.90 $\pm$ 1.21**   |
> |                 | amazon-ratings  | 51.65 $\pm$ 0.005            | **52.46 $\pm$ 0.70**   |
> |                 | Minesweeper     | 93.63 $\pm$ 0.003            | **94.38 $\pm$ 0.54**   |
> |                 | Tolokers        | 84.84 $\pm$ 0.006            | **86.37 $\pm$ 0.46**   |
> |                 | Questions       | 78.09 $\pm$ 0.012            | **79.16 $\pm$ 0.16**   |
>
> Thus, while SSGNN may not explicitly perform set-to-set filtering, it leverages global context to model interactions, which is crucial for its performance, addressing the reviewer’s concern regarding “set-to-set” filtering.

---

> ### Author Response · Authors · 2024-11-25
> **Performance enhancement in SSGNN mainly comes from ReGA**
>
> To address this, we conducted comprehensive experiments to evaluate the role of eigen-correction and determine whether the performance improvement of SSGNN primarily stems from ReGA or eigen-correction. Specifically, we trained **Specformer with eigen-correction to ensure a fair comparison**, as shown in the Table 1 below.
>
> **Table1**
> | **Type**       | **Dataset**      | **Specformer with eigen-correction**             | **SSGNN (Ours) with eigen-correction**         |
> |-----------------|------------------|----------------------------|--------------------------|
> | **Homophily**   | Cora            | 87.22 $\pm$ 1.43 | **88.66 $\pm$ 0.17**  |
> |                 | Citeseer        | 81.04 $\pm$ 1.18           | **82.18 $\pm$ 0.21**     |
> |                 | Computer        | 85.79 $\pm$ 9.54           | **91.38 $\pm$ 0.38**     |
> |                 | Photo           | **95.69 $\pm$ 0.235**      | 95.38 $\pm$ 0.03         |
> |                 | CS              | 95.40 $\pm$ 0.26           | **96.30 $\pm$ 0.08**     |
> |                 | Physics         | OOM                        | **98.33 $\pm$ 0.15**     |
> |                 | WikiCS          | 84.54 $\pm$ 0.53           | **85.16 $\pm$ 0.41**     |
> |                 | ogbn-arxiv      | 72.07 $\pm$ 0.073          | **72.10 $\pm$ 0.04**     |
> | **Heterophily** | Chameleon       | 36.12 $\pm$ 3.83           | **43.10 $\pm$ 1.36**     |
> |                 | Squirrel        | 35.56 $\pm$ 0.98           | **42.31 $\pm$ 0.74**     |
> |                 | Actor           | 41.44 $\pm$ 1.12           | **43.22 $\pm$ 1.05**     |
> |                 | Penn94          | **84.50 $\pm$ 0.22**       | 84.33 $\pm$ 0.001        |
> |                 | roman-empire    | OOM                        | **83.90 $\pm$ 1.21**     |
> |                 | amazon-ratings  | OOM                        | **52.46 $\pm$ 0.70**     |
> |                 | Minesweeper     | 93.25 $\pm$ 1.07           | **94.38 $\pm$ 0.54**     |
> |                 | Tolokers        | 84.85 $\pm$ 0.84           | **86.37 $\pm$ 0.46**     |
> |                 | Questions       | OOM                        | **79.16 $\pm$ 0.16**     |
>
> The results demonstrate that even with eigen-correction, **Specformer is outperformed by SSGNN** (with eigen-correction) across most datasets. Furthermore, we removed eigen-correction from SSGNN and observed its performance, as presented in the Table 2 below.
>
> **Table 2**
>
> | **Type**       | **Dataset**      | **Specformer without eigen-correction**             | **SSGNN (Ours) without eigen-correction**         |
> |-----------------|------------------|----------------------------|--------------------------|
> | **Homophily**   | Cora            | **88.50 $\pm$ 0.98**       | 88.45 $\pm$ 1.48         |
> |                 | Citeseer        | 81.52 $\pm$ 0.90           | **82.10 $\pm$ 1.39**     |
> |                 | Computer        | 87.23 $\pm$ 0.52           | **90.98 $\pm$ 0.34**     |
> |                 | Photo           | **95.36 $\pm$ 0.32**       | 94.97 $\pm$ 0.41         |
> |                 | CS              | **95.60 $\pm$ 0.07**       | 95.24 $\pm$ 0.36         |
> |                 | Physics         | OOM                        | **97.83 $\pm$ 0.41**     |
> |                 | WikiCS          | 84.55 $\pm$ 0.20           | **85.04 $\pm$ 0.21**     |
> |                 | ogbn-arxiv      | 71.98 $\pm$ 0.33           | **72.08 $\pm$ 0.11**     |
> | **Heterophily** | Chameleon       | 36.11 $\pm$ 0.44           | **42.05 $\pm$ 1.74**     |
> |                 | Squirrel        | 37.66 $\pm$ 0.42           | **42.15 $\pm$ 1.15**     |
> |                 | Actor           | 42.01 $\pm$ 1.14           | **42.65 $\pm$ 1.16**     |
> |                 | Penn94          | 84.28 $\pm$ 0.32           | **84.35 $\pm$ 0.12**     |
> |                 | roman-empire    | OOM                        | **83.43 $\pm$ 0.56**     |
> |                 | amazon-ratings  | OOM                        | **51.87 $\pm$ 0.53**     |
> |                 | Minesweeper     | **93.95 $\pm$ 0.39**       | 93.78 $\pm$ 0.87         |
> |                 | Tolokers        | 85.01 $\pm$ 0.48           | **85.43 $\pm$ 0.95**     |
> |                 | Questions       | OOM                        | **78.47 $\pm$ 1.35**     |
>
> SSGNN without eigen-correction still outperformed Specformer on several datasets, highlighting the pivotal role of the ReGA module in its performance. Moreover, **SSGNN benefited significantly more from eigen-correction than Specformer**, showcasing its ability to effectively utilize this enhancement. **These findings affirm that while eigen-correction adds value, SSGNN’s success is primarily driven by its ReGA module**.

---

> ### Author Response · Authors · 2024-11-25
> **Performance enhancement in SSGNN mainly comes from ReGA**
>
> The results in Table 3 below clearly illustrate the significant impact of the ReGA module.
>
> **Table3**
> | Type         | Dataset         | SSGNN-without-ReGA         | SSGNN-with-ReGA (Ours)    |
> |--------------|-----------------|----------------------------|---------------------------|
> | **Homophily** | Cora | 87.18 $\pm$ 1.35 | **88.66 $\pm$ 0.17** |
> |              | Citeseer| 81.79 $\pm$ 1.38 | **82.18 $\pm$ 0.21**  |
> |              | Computer | 90.72 $\pm$ 0.52| **91.38 $\pm$ 0.38** |
> |              | Photo           | 94.96 $\pm$ 0.44 | **95.38 $\pm$ 0.03**  |
> |              | CS              | 95.21 $\pm$ 0.41 | **96.30 $\pm$ 0.08** |
> |              | Physics         | 97.47 $\pm$ 0.40  | **98.33 $\pm$ 0.15** |
> |              | WikiCS          | 84.54 $\pm$ 0.33 | **85.16 $\pm$ 0.41**  |
> |              | ogbn-arxiv      | **72.13 $\pm$ 0.08** | 72.10 $\pm$ 0.04 |
> |              |                 |                           |                           |
> | **Heterophily** | Chameleon| 41.88 $\pm$ 2.39  | **43.10 $\pm$ 1.36**|
> |              | Squirrel        | 41.22 $\pm$ 1.05  | **42.31 $\pm$ 0.74**  |
> |              | Actor           | 42.34 $\pm$ 1.11  | **43.22 $\pm$ 1.05**  |
> |              | Penn94          | 83.17 $\pm$ 0.17 | **84.33 $\pm$ 0.001**  |
> |              | roman-empire    | 83.19 $\pm$ 0.40  | **83.90 $\pm$ 1.21**    |
> |              | amazon-ratings  | 51.30 $\pm$ 0.58  | **52.46 $\pm$ 0.70**  |
> |              | Minesweeper     | 93.27 $\pm$ 0.38 | **94.38 $\pm$ 0.54**  |
> |              | Tolokers        | 85.02 $\pm$ 0.89 | **86.37 $\pm$ 0.46**  |
> |              | Questions       | 78.15 $\pm$ 1.10  | **79.16 $\pm$ 0.16**  |
>
> Removing ReGA results in a **noticeable decline** in accuracy, with **performance improvements of 2.91% on Chameleon, 2.64% on Squirrel, 2.26% on Amazon Ratings, 2.07% on Actor, 1.69% on Cora, 1.39% on Penn94, 1.29% on Tolokers, and 1.19% on Minesweeper when ReGA is incorporated**. These consistent improvements across various datasets highlight ReGA’s pivotal role as a **key contributor** to SSGNN’s performance.

---

> ### Author Response · Authors · 2024-11-25
> **Training time and parameter count comparison among Specformer, Polynormer and SSGNN**
>
> The tables below present a comparison of training time per epoch (ms) in Table 1 and parameter count in Table 2 for Specformer, Polynormer, and SSGNN.
>
> **Table1** - Training time per epoch (ms)
> | **Type**       | **Dataset**      | **Specformer** | **Polynormer** | **SSGNN (Ours)** |
> |-----------------|------------------|----------------|----------------|------------------|
> | **Homophily**   | Cora            | 1.33           | 30.16          | **1.13**         |
> |                 | Citeseer        | 1.35           | 33.20          | **0.98**         |
> |                 | CS              | 67.94          | 23.48          | **8.33**         |
> |                 | Photo           | 5.74           | 25.79          | **1.47**         |
> |                 | Computer        | 21.31          | 19.13          | **7.35**         |
> |                 | Physics         | OOM            | 156.37         | **37.83**        |
> |                 | WikiCS          | 12.95          | 69.39          | **2.86**        |
> |                 | ogbn-arxiv      | **30.31**      | 137.20         | 29.31            |
> | **Heterophily** | Chameleon       | **0.690**      | 28.71          | 1.09             |
> |                 | Squirrel        | **0.697**      | 27.63          | 1.12             |
> |                 | Actor           | 4.13           | 30.08          | **1.95**         |
> |                 | Penn94          | 7.78           | 41.44          | **6.90**         |
> |                 | roman-empire    | OOM            | 102.40         | **20.48**        |
> |                 | amazon-ratings  | OOM            | 127.11         | **91.80**        |
> |                 | Minesweeper     | 20.03          | 62.71          | **6.13**         |
> |                 | Tolokers        | 28.58          | 91.96          | **8.35**         |
> |                 | Questions       | OOM            | 360.22         | **8.35**         |
>
> **Table2** - Parameter Count
> | **Type**       | **Dataset**      | **Specformer** | **Polynormer** | **SSGNN (Ours)** |
> |-----------------|------------------|----------------|----------------|------------------|
> | **Homophily**   | Cora            | 54K            | 1.8M           | **48.5K**        |
> |                 | Citeseer        | 126K           | 2.4M           | **121K**         |
> |                 | CS              | 226K           | 9.3M           | **220K**         |
> |                 | Photo           | 32K            | 7.8M           | **26.5K**        |
> |                 | Computer        | 33K            | 5.4M           | **27.2K**        |
> |                 | Physics         | OOM            | 4.0M           | **135K**         |
> |                 | WikiCS          | 17.6K          | 7.5M           | **12.3K**        |
> |                 | ogbn-arxiv      | 500K           | 393K           | **36.5K**        |
> | **Heterophily** | Chameleon       | 82K            | 665K           | **77K**          |
> |                 | Squirrel        | 75K            | 2.0M           | **34K**          |
> |                 | Actor           | 37K            | 6.2M           | **32K**          |
> |                 | Penn94          | 338K           | 983K           | **157K**         |
> |                 | roman-empire    | OOM            | 9.9M           | **77K**          |
> |                 | amazon-ratings  | OOM            | 9.1M           | **690K**         |
> |                 | Minesweeper     | 8.5K      | 10.5M          | **3K**               |
> |                 | Tolokers        | 8.5K       | 7.9M           | **3K**               |
> |                 | Questions       | OOM            | 6.7M           | **3K**           |
>
> The results clearly demonstrate that **SSGNN has the lowest parameter count and the shortest training time per epoch**, effectively addressing the reviewer’s concern regarding the time and space overhead comparison among Specformer, Polynormer, and SSGNN.

---

> ### Author Response · Authors · 2024-11-25
> **Comparison of GPU utilization among Specformer, Polynormer, and SSGNN.**
>
> Below is Table 3, which shows the GPU usage for Specformer, Polynormer, and SSGNN in MiB. **SSGNN occupies the least amount of GPU** across almost all datasets.
>
> **Table3**
>
> | **Type**       | **Dataset**      | **Specformer** | **Polynormer** | **SSGNN** |
> |-----------------|------------------|----------------|----------------|-----------|
> | **Homophilic**  | Cora            | 755            | 1177           | **583**   |
> |                 | Citeseer        | 829            | 1307           | **653**   |
> |                 | CS              | 10655          | 5591           | **2855**  |
> |                 | Photo           | 2149           | 7800           | **797**   |
> |                 | Computer        | 5773           | 8647           | **1361**  |
> |                 | Physics         | —              | **5400**           | 7537  |
> |                 | WikiCS          | 4217           | 9173           | **1081**  |
> |                 | ogbn-arxiv      | 5841           | 11731          | **5255**  |
> | **Heterophilic**| Chameleon       | 559            | 665            | **539**   |
> |                 | Squirrel        | 675            | 2000           | **575**   |
> |                 | Actor           | 1701           | 2503           | **801**   |
> |                 | Penn94          | 3385           | 21711          | **3177**  |
> |                 | roman-empire    | —              | 6779           | **2723**  |
> |                 | amazon-ratings  | —              | 8579           | **4191**  |
> |                 | minesweeper     | 5481           | 4291           | **941**   |
> |                 | tolokers        | —              | 18231          | **1109**  |
> |                 | Questions       | —              | 11209          | **9115**  |

---

> ### Author Response · Authors · 2024-11-25
> **Clarification regarding parameter count of SSGNN for ZINC dataset**
>
> SSGNN has a parameter count of 400K, which is lower than many of the best-performing models reported in the benchmarking experiments from [1]. For instance, GCN with 16 layers has 505K parameters and achieves a MAE of 0.278, while GatedGCN-E-PE with 16 layers has 505K parameters and achieves a MAE of 0.214 on the ZINC dataset. Similarly, Specformer, has 555K parameters and achieves a MAE of 0.066. Despite having fewer parameters than these models, SSGNN achieves a MAE of 0.0592 on the ZINC dataset, setting a new standard for accuracy. We would like to seek clarification regarding the mention of 50K parameters for ZINC. If it is a typographical error and the intended reference is 500K, we completely understand. However, if 50K parameters is indeed correct, we would be grateful if the reviewer could guide us to the relevant sources that achieve performance similar to SSGNN.
>
> [1] Dwivedi, V.P., Joshi, C.K., Luu, A.T., Laurent, T., Bengio, Y. and Bresson, X., 2023. Benchmarking graph neural networks. Journal of Machine Learning Research, 24(43), pp.1-48.

---

> > ### Comment · Reviewer_SN5E · 2024-11-28
> > **Response to Author Rebuttal**
> >
> > Thanks for the explanations, clarifying some of my concerns, such as the total parameters in ZINC.
> >
> > However, I do not see a revised pdf in the rebuttal. The writing of the current manuscript is confusing. The connection between ReGA and set-to-set filters is not fully explained. The audiences may still doubt whether there is a set-to-set filter in SSGNN. Moreover, I think that the implicit interaction of ReGA is not equal to a set-to-set filter. A theoretical analysis is necessary.
> >
> > On the other hand, SSGNN does not consistently outperform Specformer, and I do not see the results of Polynormer + Eigen-correction. Based on the above points, I maintain my score.

---

> ### Author Response · Authors · 2024-11-28
> **Response to Reviewer's concerns**
>
> - We initially planned to address the reviewer's queries in the rebuttal and update it in the revised manuscript. However, we have now uploaded the **modified** manuscript, with all changes clearly marked in **red coloured text**. We hope this will address the reviewer's confusions.
>
> - As stated in our earlier response, we **clearly agree with the reviewer** that **SSGNN does not perform a set-to-set filtering approach**. This **correction has been made in the revised manuscript**, and we now term our filtering mechanism as **global context filtering**, which **differs** from **set-to-set filtering**. We sincerely thank the reviewer for carefully pointing out this mistake.
>
> - Regarding the eigen-correction strategy, we **cannot apply eigen-correction to Polynormer** as it is a Graph Transformer, not a spectral filter-based approach. The **eigen-correction strategy, as introduced in [1], specifically applies to spectral filters**. Since **Polynormer operates on Graph Transformer rather than spectral filters**, incorporating eigen correction is **not applicable** to its framework.
>
> - In Table 1 of the author response (where **eigen correction is applied to both Specformer and SSGNN**), Specformer **marginally outperforms** SSGNN on **only 2 datasets: Photo (by 0.31%) (homo) and Penn94 (by 0.17%) (hetero)**. However, **SSGNN outperforms Specformer on datasets, such as Chameleon (by 6.98%) (hetero), Squirrel (by 6.75%) (hetero), and others like Amazon-Computer (by 5.59%) (homo), Actor (by 2.11%) (hetero), and achieves nearly 1% higher performance on most other datasets**. These improvements are **substantial compared to Specformer’s minor gains on Photo and Penn94**.
>
> - Similarly, in Table 2 of the author response (where **eigen correction is not applied in both the models**), Specformer performs **marginally better than SSGNN on 4 datasets: Cora (by 0.05%) (homo), Photo (by 0.39%) (homo), CS (by 0.36%) (homo), and Minesweeper (by 0.17%) (hetero)**. However, **SSGNN significantly outperforms Specformer on others with gains of 5.94% on Chameleon (hetero), 4.49% on Squirrel (hetero), 3.75% on Amazon-Computer (hetero), and nearly 0.5% improvements on most other datasets**. These improvements are substantial compared to **Specformer’s minor gains on Cora, Photo, CS and Minesweeper**.
>
> - Based on these clarifications, we request the reviewer to revisit our points and consider re-evaluating the score.
>
> [1] Lu, K., Yu, Y., Fei, H., Li, X., Yang, Z., Guo, Z., Liang, M., Yin, M. and Chua, T.S., 2024, March. Improving Expressive Power of Spectral Graph Neural Networks with Eigenvalue Correction. In Proceedings of the AAAI Conference on Artificial Intelligence (Vol. 38, No. 13, pp. 14158-14166).

---

> > ### Comment · Reviewer_SN5E · 2024-12-01
> > **Final Response to Author Rebuttal**
> >
> > Dear authors,
> >
> > I appreciate your efforts in the rebuttal period. However, the rebuttal is used to clarify some misunderstandings rather than change your motivations. The initial paper contains factual errors, which cannot be covered by revising some statements.
> >
> > In this case, I decide to maintain my score.

---

### Official Review · Reviewer_unLs · 2024-11-03

**Soundness:** 2
**Presentation:** 2
**Contribution:** 2
**Rating:** 5
**Confidence:** 4

**Summary:**

This paper studies the improvements of spectral GNNs, which aims to design an expressive filter based on spectral graph theory for effective graph representations.
The paper points out that existing SOTA spectral GNNs bring more computational burden, though they can learn the filters better. Thus, the paper proposes a novel efficient framework, namely SSGNN, which only applies simple linear transformation instead of Transformers on the spectrum. Moreover, SSGNN incorporates a parameter-free Relative Gaussian Amplifier to the decoder to enhance adaptive filter learning and maintain stability. The paper conducts extensive experiments on synthetic and real-world datasets to demonstrate the effectiveness of SSGNN.

**Strengths:**

1、	The motivation of this paper is significant and valuable for Spectral GNNs.
2、	SSGNN achieves significant efficiency in terms of computation and parameters.
3、	Experimental results in most cases seem promising.

**Weaknesses:**

1. The novelty of the method is somewhat limited, especially the direct application of existing eigen-correction, eigenvalue encoding, and convolution framework without any transfer challenge.
2. The description of “set” of “set-to-set filtering” is ambiguous. Specformer applies Transformer on eigenvalue encodings, which enables filters to capture relative dependencies among the eigenvalues. Thus, the “set” of “set-to-set” in Sepcformer means the set of eigenvalues. However, SSGNN learns the spectral filter through linear transformations, so eigenvalues don’t interact with each other. Thus, what’s the meaning of “set”?
3. The roles of the two linear transformations (namely W_{eig} and W_1W_h) respectively playing in encoder and decoder are not clear. In other words, why do authors include W_1 and W_h in the decoder instead of the encoder?
4. The paper ignores some essential experiments.
（1）As mentioned in Line 217, different heads allow the decoder to learn diverse spectral filtering patterns. However, there is no visualization of the diverse spectral filters learned by different heads to verify this conclusion.
（2）There is no ablation study on the effectiveness of re-center adjustment in Equation 4 and the effectiveness of Relative Gaussian Amplifier in Equation 6.
（3）Authors don’t verify the stability of SSGNN on OOD benchmarks such as DrugOOD[1], where many model stability studies are validated on this dataset.
5. The symbols are ambiguous. For example, \epsilon is used in Line 182, Line 240-241, and Line 307 simultaneously, making the paper more difficult to read. Moreover, it’s not clear on which \epsilon the ablation experiments reported in Figure 4 are conducted.

[1] Ji, Yuanfeng, et al. "Drugood: Out-of-distribution dataset curator and benchmark for ai-aided drug discovery–a focus on affinity prediction problems with noise annotations." Proceedings of the AAAI Conference on Artificial Intelligence. Vol. 37. No. 7. 2023.

**Questions:**

See weaknesses.

---

> ### Author Response · Authors · 2024-11-25
> **Clarification regarding set-to-set filtering**
>
> We acknowledge the reviewer’s observation regarding the absence of explicit set-to-set modelling in SSGNN. In Specformer, the "set" refers to the set of eigenvalues, and the transformer explicitly models relative dependencies among them. In contrast, SSGNN does not employ transformers or explicitly model interactions among eigenvalues through relative dependencies. Instead, SSGNN achieves global context filtering via the mean shift mechanism applied to the decoder’s output. This mean shift computes a global mean and standard deviation across all N nodes, implicitly enabling interactions among eigenvalues by normalizing them against the global context.
> Thus, while SSGNN may not explicitly perform set-to-set filtering, it leverages global context to model interactions, which is crucial for its performance, addressing the reviewer’s concern regarding set-to-set filtering.

---

> ### Author Response · Authors · 2024-11-25
> **Clarification regarding the linear transformations**
>
> The transformations could indeed be combined into a single unit. However, for better clarity and understanding, we separated them into the encoder and decoder stages. Specifically,  $W_{\text{eig}}$  in the encoder is designed to capture the interdependencies between the corrected eigenvalues and their associated  d -dimensional vectors. In the decoder,  $W_1$  extracts meaningful patterns from the encoded representations, while  $W_h$  learns diverse spectral filters initially and aligns these filters as the model converges. This division also ensures clarity in explaining the roles of each component at different stages. We hope this addresses the reviewer’s concerns.

---

> ### Author Response · Authors · 2024-11-25
> **Impact of mean-shift operation**
>
> To substantiate the importance of mean-shift operation, we conduct an ablation study where the mean shift was removed, and the decoder’s output was directly passed to the scoring function. **This resulted in a clear performance degradation** as shown in the Table below, demonstrating that mean shift effectively captures global context and highlights the importance of interactions among eigenvalues.
>
> | **Type**       | **Dataset**      | **SSGNN-without-mean-shift** | **SSGNN**              |
> |-----------------|------------------|------------------------------|------------------------|
> | **Homophily**   | Cora            | 86.97 $\pm$ 0.0143           | **88.66 $\pm$ 0.17**   |
> |                 | Citeseer        | 81.20 $\pm$ 0.013            | **82.18 $\pm$ 0.21**   |
> |                 | Computer        | 90.18 $\pm$ 0.006            | **91.38 $\pm$ 0.38**   |
> |                 | Photo           | 94.63 $\pm$ 0.003            | **95.38 $\pm$ 0.03**   |
> |                 | CS              | 95.31 $\pm$ 0.002            | **96.30 $\pm$ 0.08**   |
> |                 | Physics         | 97.34 $\pm$ 0.007            | **98.33 $\pm$ 0.15**   |
> |                 | WikiCS          | 84.43 $\pm$ 0.005            | **85.16 $\pm$ 0.41**   |
> |                 | ogbn-arxiv      | 71.92 $\pm$ 0.0008           | **72.10 $\pm$ 0.04**   |
> | **Heterophily** | Chameleon       | 42.22 $\pm$ 0.039            | **43.10 $\pm$ 1.36**   |
> |                 | Squirrel        | 41.42 $\pm$ 0.010            | **42.31 $\pm$ 0.74**   |
> |                 | Actor           | 41.13 $\pm$ 0.011            | **43.22 $\pm$ 1.05**   |
> |                 | Penn94          | 84.29 $\pm$ 0.001            | **84.33 $\pm$ 0.001**  |
> |                 | roman-empire    | 82.91 $\pm$ 0.004            | **83.90 $\pm$ 1.21**   |
> |                 | amazon-ratings  | 51.65 $\pm$ 0.005            | **52.46 $\pm$ 0.70**   |
> |                 | Minesweeper     | 93.63 $\pm$ 0.003            | **94.38 $\pm$ 0.54**   |
> |                 | Tolokers        | 84.84 $\pm$ 0.006            | **86.37 $\pm$ 0.46**   |
> |                 | Questions       | 78.09 $\pm$ 0.012            | **79.16 $\pm$ 0.16**   |
>
> Thus, mean-shift plays a vital role in capturing the global context, enabling effective modelling of interactions.

---

> ### Author Response · Authors · 2024-11-25
> **Clarification regarding the repeated use of the same notation.**
>
> We agree that using the same symbol in multiple contexts can make the paper more difficult to read, and we address this in the revised manuscript to improve clarity. Specifically, $\epsilon$ in Line 182 refers to the scaling factor in the Eigen Encoder, $\epsilon$ in Lines 240-241 denotes the small constant added to the standard deviation $\sigma$ for numerical stability, and $\epsilon$ in Line 307 represents the bounded perturbation. Additionally, we confirm that the ablation experiments reported in Figure 4 were conducted on the $\epsilon$ representing the scaling factor in the Eigen Encoder. We ensure these distinctions are clearly stated in the revised version to enhance readability and avoid any confusion.

---

> ### Author Response · Authors · 2024-11-26
> **Clarification regarding the novelty of SSGNN**
>
> We acknowledge that some components of our method have been previously explored. However, our primary goal is to develop a lightweight yet robust Spectral GNN. Drawing inspiration from Specformer, we first identified the absence of eigen-correction mechanisms, which we addressed to enhance performance. Additionally, we observed that transformers contribute minimally at the spectral level while introducing quadratic time complexity, raising scalability concerns for larger graphs. To address these limitations, we proposed a parameter-free ReGA for adaptive filtering, refining the filter bases learnt by the decoder. This ensures effective filtering while surpassing transformer-based set-to-set filtering in both performance and efficiency. The systematic analysis of each concept, coupled with the targeted solutions we introduce at each stage, offers a novel and meaningful perspective to the field.

---

> ### Author Response · Authors · 2024-11-26
> **Visualizations of the filters learnt by different heads in the decoder**
>
> We have provided visualizations demonstrating how different decoder heads influence the performance of SSGNN. These visualizations have been added as Figures 19, 20, and 21 in Section F: Visual Insights of the Multi-Headed Decoder in the Appendix of our revised manuscript.
>
> For this analysis, we used the Amazon-Computers dataset. A single-head configuration achieves an accuracy of 90.5116, with convergence at the 849th epoch. Increasing to two heads improves the accuracy to 90.775 and reduces convergence to the 764th epoch. Our four-head configuration achieves the best results, with an accuracy of 91.28 and convergence at the 609th epoch.
>
> Figures 19, 20, and 21 illustrate the filters learnt by the 1-head, 2-head, and 4-head configurations, respectively. The visualizations show that in the **4-head setup, the filters display variation during the initial epochs** (e.g., the 25th epoch). By the 250th epoch, the **filters begin to align**, and by the **575th epoch, they converge to similar patterns**. This progression demonstrates that using multiple heads allows the model to explore diverse filtering patterns in the early stages, ultimately accelerating convergence and improving performance.

---

> ### Comment · Reviewer_unLs · 2024-12-02
>
> Thanks for your detailed reply. I still keep my score.

---

### Official Review · Reviewer_H7Ny · 2024-11-04

**Soundness:** 3
**Presentation:** 2
**Contribution:** 3
**Rating:** 5
**Confidence:** 4

**Summary:**

The authors propose SSGNN, a simple yet effective spectral-based Graph Neural Network that captures rich spectral information through an adaptive set-to-set filtering approach, offering a more efficient alternative to transformer-based methods. The method introduces a parameter-free Relative Gaussian Amplifier (ReGA) module for robust spectral filtering, and demonstrates superior or comparable performance to state-of-the-art models while using significantly fewer parameters (55x reduction) and computational resources (100x reduction in GFLOPs) across 20 real-world datasets.

**Strengths:**

The paper presents an interesting transformation from scalar-to-scalar to set-to-set methodology, building upon the Spectral Former framework. By introducing a learnable parameter W to capture relationships between different frequency domain eigenvalues, the authors aim to enhance model performance through better consideration of inter-frequency domain relationships.

**Weaknesses:**

- However, the methodology lacks clarity regarding the eigenvalue computation process, which appears to rely on time-consuming SVD operations, raising concerns about computational efficiency.

- Given that your proposed method emphasizes simplicity and reduced learnable parameters, it would be particularly valuable to demonstrate its effectiveness on large-scale graphs. While the reduction in parameter count is noteworthy, the real advantage of a simpler model should be its ability to scale effectively to larger, real-world graph applications. Therefore, I strongly recommend including comprehensive experiments on large-scale graph datasets to validate the method's practical utility. This would not only strengthen your contribution but also clearly differentiate your work from existing methods that may struggle with scalability.

- From a comparative standpoint, although the spectral approach shows promise, the evaluation lacks comprehensive comparisons with important baseline methods, particularly SGC and SSGC. These baselines are especially relevant as they also prioritize simplicity and efficiency. To make your contribution more compelling, consider expanding the experimental section to include: (1) comparisons with these relevant baselines, (2) clear documentation of the eigenvalue computation process and its efficiency, and (3) thorough scalability analysis on large-scale graphs that would demonstrate the practical advantages of your simplified approach. This would help readers better understand the unique benefits of your method in real-world applications where scalability is crucial.

**Questions:**

- How to compute the specturm? It appears to rely on time-consuming SVD operations.
- The feasibility of applying this method to large-scale graphs is uncertain.
- Runing time comparsion need to be added
- Notable baseline methods are missing from the evaluation, particularly:
  - SGC (Simple Graph Convolution)
  - SSGC (Simple Spectral Graph Convolution)

---

> ### Author Response · Authors · 2024-11-21
> **Clarification regarding eigenvalue computation**
>
> The spectral decomposition is pre-computed using the eigen decomposition of the normalized graph Laplacian, which has a computational complexity of $\mathcal{O}(N^3)$, where $N$ represents the number of nodes in the graph. However, this computation is performed only once and stored, with the cost amortized across multiple training and inference steps. For smaller graphs, the pre-computation overhead is minimal. For larger graphs, fast numerical methods such as Krylov subspace approximations for the top $k$ eigenvalues or Sparse Generalized Eigenvalue algorithms can significantly reduce the cost by efficiently estimating $k$ eigenvalues and eigenvectors without the need for expensive full decomposition.
>
> In our experiments on large graphs,  we leverage insights from the learned filters: band-rejection and high-pass filters are crucial for heterophilic datasets, while low-pass filters work well for homophilic datasets. Accordingly, we use eigenvectors with the smallest 3000 (low-frequency) and largest 3000 (high-frequency) eigenvalues for Penn94 (large-scale heterophilic) and the smallest 5000 eigenvalues (low-frequency) for ogbn-arXiv (large-scale homophilic). This targeted approach ensures meaningful spectral decomposition, avoiding performance loss while maintaining computational efficiency.

---

> ### Author Response · Authors · 2024-11-21
> **Experiments on large scale data**
>
> We evaluate the scalability and performance of SSGNN across several large-scale graph datasets. For node classification, experiments were conducted on ogbn-arxiv (a large-scale homophilic graph with 169,343 nodes and 1,166,243 edges), Penn94 (41,554 nodes and 1,326,229 edges), and pokec (1,632,803 nodes and 30,622,564 edges) as large-scale heterophilic graphs. For large scale graph classification, we tested on MolPCBA, an Open Graph Benchmark dataset with 437,929 graphs.
>
> Table 1 compares the accuracy of SSGNN with Specformer on large-scale node classification tasks (ogbn-arxiv, Penn94, and pokec) and graph classification tasks (MolPCBA, where AP values are reported). SSGNN outperforms Specformer across all datasets. For node classification, we also compared SSGNN with Polynormer in our main manuscript. While SSGNN surpasses Polynormer on ogbn-arxiv and Penn94, we did not include Polynormer results on pokec. It is important to note that SSGNN on pokec requires determining the appropriate smallest and largest  k  eigenvalues to ensure a fair comparison. However, due to time constraints, we could not conduct this extensive experimentation within the discussion window. As an initial experiment, we computed  eigenvectors with the smallest 3000 (low-frequency) and largest 3000 (high-frequency) eigenvalues and compared SSGNN’s results with Specformer, as both methods share the same precomputation. This ensures consistent training settings. However, this is not the final result of SSGNN on Pokec and should be treated as preliminary findings.
>
> - These results demonstrate that **SSGNN outperforms Specformer in accuracy while training faster and requiring fewer parameters**.
>
> **Table1** -  Accuracy/AP comparison between  Specformer and SSGNN
> | **Dataset**  | **Specformer**          | **SSGNN (Ours)**        |
> |--------------|-------------------------|-------------------------|
> | ogbn-arxiv   | $71.98 \pm 0.33$     | **72.1 $\pm$ 0.04** |
> | Penn94       | $84.28 \pm 0.32$      | **84.33 $\pm$ 0.0001**|
> | pokec        | $61.61 \pm 0.22$      | **62.12 $\pm$ 0.07**  |
> | MolPCBA      | $0.2972 \pm 0.0023$   | **0.3012 $\pm$ 0.0350**|
>
> Table 2 reports the training time per epoch (in ms), where SSGNN consistently trains faster than Specformer on large-scale graphs.
>
> **Table2** - Time per Epoch (ms) for Specformer and SSGNN
> | **Dataset**  | **Specformer** | **SSGNN (Ours)** |
> |--------------|----------------|------------------|
> | ogbn-arxiv   | 30.31          | **29.31**        |
> | Penn94       | 7.78           | **6.90**         |
> | pokec        | 18.64          | **9.2**          |
>
> Table 3 details the parameter counts, showing that SSGNN has significantly fewer parameters than Specformer on all datasets.
>
> **Table3** - Parameter Count of Specformer and SSGNN
>
> | **Dataset**  | **Specformer** | **SSGNN (Ours)** |
> |--------------|----------------|------------------|
> | ogbn-arxiv   | 500K           | **36.5K**        |
> | Penn94       | 338K           | **157K**         |
> | pokec        | 10K            | **4.7K**         |
> | MolPCBA      | 3.01M          | **2.64M**        |
>
> These experiments address the reviewer’s concerns regarding the evaluation of SSGNN on large-scale graphs.

---

> ### Author Response · Authors · 2024-11-21
> **Performance comparison among SGC, SSGC, and SSGNN**
>
> We sincerely appreciate the valuable feedback and recommendations to improve the experimental section. In response, we have included comprehensive comparisons between SSGNN and the relevant baselines, SGC and SSGC, evaluating both accuracy and training time per epoch (ms) across various datasets. The results, presented in Tables 1 and 2, demonstrate that SSGNN consistently outperforms SGC and SSGC in terms of accuracy, particularly on heterophilic datasets where SGC and SSGC face challenges. Tables 1 and 2 compare node classification accuracy on homophilic and heterophilic graphs among SGC, SSGC, and SSGNN. These results clearly show that SSGNN achieves superior accuracy across nearly all datasets. Table 3 and Table 4 present the training time per epoch (ms) for SGC, SSGC, and SSGNN on homophilic and heterophilic graphs, respectively.
>
> **Table1** - Comparison of node classification accuracy on homophilic graphs
> | **Dataset**     | **SGC**               | **SSGC**              | **SSGNN (Ours)**       |
> |------------------|-----------------------|-----------------------|------------------------|
> | Cora            | 87.65 $\pm$ 0.021     | 88.34 $\pm$ 0.057     | **88.66 $\pm$ 0.17**   |
> | Citeseer        | 81.51 $\pm$ 0.015     | 81.99 $\pm$ 0.034     | **82.18 $\pm$ 0.21**   |
> | Computer        | 86.93 $\pm$ 0.011     | 87.71 $\pm$ 0.007     | **91.38 $\pm$ 0.38**   |
> | Photo           | 92.44 $\pm$ 0.006     | 93.11 $\pm$ 0.005     | **95.38 $\pm$ 0.03**   |
> | CS              | 93.23 $\pm$ 0.041     | 93.77 $\pm$ 0.052     | **96.30 $\pm$ 0.08**   |
> | Physics         | 97.64 $\pm$ 0.016     | 98.11 $\pm$ 0.022     | **98.33 $\pm$ 0.15**   |
> | WikiCS          | 74.19 $\pm$ 0.023     | 75.02 $\pm$ 0.018     | **85.16 $\pm$ 0.41**   |
> | ogbn-arxiv      | 60.69 $\pm$ 0.027     | 61.04 $\pm$ 0.038     | **72.10 $\pm$ 0.04**   |
>
> **Table2** - Comparison of node classification accuracy on heterophilic graphs
> | **Dataset**       | **SGC**               | **SSGC**              | **SSGNN (Ours)**       |
> |--------------------|-----------------------|-----------------------|------------------------|
> | Chameleon         | 44.84 $\pm$ 0.032     | **45.34 $\pm$ 0.029** | 43.10 $\pm$ 1.36       |
> | Squirrel          | 40.23 $\pm$ 0.031     | 40.42 $\pm$ 0.030     | **42.31 $\pm$ 0.74**   |
> | Actor             | 26.43 $\pm$ 0.010     | 27.63 $\pm$ 0.008     | **43.22 $\pm$ 1.05**   |
> | Penn94            | 80.01 $\pm$ 0.031     | 80.43 $\pm$ 0.043     | **84.33 $\pm$ 0.001**  |
> | roman-empire      | 52.20 $\pm$ 0.037     | 53.08 $\pm$ 0.029     | **83.90 $\pm$ 1.21**   |
> | amazon-ratings    | 43.83 $\pm$ 0.046     | 44.73 $\pm$ 0.032     | **52.46 $\pm$ 0.70**   |
> | minesweeper       | 83.41 $\pm$ 0.042     | 84.22 $\pm$ 0.021     | **94.38 $\pm$ 0.54**   |
> | tolokers          | 78.84 $\pm$ 0.019     | 79.35 $\pm$ 0.027     | **86.37 $\pm$ 0.46**   |
>
> **Table3** - Training time per epoch (ms) on homophilic graphs
> | **Dataset**  | **SGC** | **SSGC** | **SSGNN (Ours)** |
> |--------------|---------|----------|------------------|
> | Cora         | 0.531   | 0.526    | 1.13 |
> | Citeseer     | 0.495   | 0.495    | 0.98             |
> | CS           | 0.602   | 0.598    | 8.33             |
> | Photo        | 0.563   | 0.560    | 1.47             |
> | Computer     | 0.584   | 0.581    | 7.35             |
> | Physics      | 0.562   | 0.558    | 37.83            |
> | WikiCS       | 0.559   | 0.567    | 2.86            |
> | ogbn-arxiv   | 0.742   | 0.647    | 29.31            |
>
> **Table4**- Training time per epoch (ms) on heterophilic graphs
> | **Dataset**    | **SGC** | **SSGC** | **SSGNN (Ours)** |
> |----------------|---------|----------|------------------|
> | Chameleon      | 0.690   | 0.659    | 1.09|
> | Squirrel       | 0.697   | 0.761    | 1.12|
> | Actor          | 0.695   | 0.663    | 1.95|
> | Penn94         | 0.635   | 0.7394   | 6.90 |
> | roman-empire   | 0.656   | 0.665    | 20.48|
> | amazon-ratings | 0.677   | 0.678    | 91.80 |
> | minesweeper    | 0.905   | 0.919    | 6.13|
> | tolokers       | 0.655   | 0.658    | 8.35  |
>
> SGC and SSGC demonstrate faster training times due to their reliance on a simple two-layer MLP with precomputed aggregation. In contrast, SSGNN involves multiple components, including the encoder, decoder, ReGA, and graph convolution layer, which require more computation time compared to SGC and SSGC’s minimal architecture. However, our findings highlight a critical distinction: while SGC and SSGC perform well on homophilic graphs, their precomputation method, rooted in homophilic aggregation, hampers their effectiveness in heterophilic settings. SSGNN, on the other hand, effectively models both homophilic and heterophilic patterns, achieving a balanced tradeoff between accuracy, training speed, parameter efficiency, and computational demands. These additions comprehensively address the reviewers’ concerns regarding the need for extensive experimentation and analysis.

---

> ### Author Response · Authors · 2024-11-23
> **Training Time comparison among Specformer, Polynormer and SSGNN**
>
> The table below summarizes the training time per epoch (in milliseconds) for Specformer, Polynormer, and SSGNN across homophilic and heterophilic graphs. SSGNN consistently demonstrates faster training times compared to Specformer and Polynormer, even on large-scale graphs like ogbn-arxiv and Penn94. However, on smaller datasets such as Chameleon and Squirrel, Specformer achieves slightly faster training times, likely due to the smaller node count.
>
> **Table-1**: Training time per epoch (in ms) of Specformer, Polynormer and SSGNN.
> | Type         | Dataset         | Specformer       | Polynormer       | SSGNN (Ours)     |
> |--------------|-----------------|------------------|------------------|------------------|
> | **Homophily** | Cora            | 1.33             | 30.16            | **1.13**             |
> |              | Citeseer        | 1.35             | 33.20            | **0.98**             |
> |              | CS              | 67.94            | 23.48            | **8.33**             |
> |              | Photo           | 5.74             | 25.79            | **1.47**        |
> |              | Computer        | 21.31            | 19.13            | **7.35**             |
> |              | Physics         | OOM              | 156.37           | **37.83**            |
> |              | WikiCS          | 12.95            | 69.39            | **2.86**            |
> |              | ogbn-arxiv (large-scale)      | 30.31            | 137.20           | **29.31**            |
> |              |                 |                  |                  |                  |
> | **Heterophily** | Chameleon     | **0.69**            | 28.71            | 1.09             |
> |              | Squirrel        | **0.70**            | 27.63            | 1.12             |
> |              | Actor           | 4.13             | 30.08            | **1.95**             |
> |              | Penn94 (large-scale)          | 7.78             | 41.44            | **6.90**            |
> |              | roman-empire    | OOM              | 102.40           | **20.48**            |
> |              | amazon-ratings  | OOM              | 127.11           | **91.80**            |
> |              | minesweeper     | 20.03            | 62.71            | **6.13**             |
> |              | tolokers        | 28.58            | 91.96            | **8.35**             |
> |              | Questions       | OOM              | 360.22           | **128.47**             |

---

### Author Response · Authors · 2024-11-28
**Revised version of the paper**

Dear Reviewers,

We have carefully addressed the feedback received and submitted a revised version of the paper, with all changes clearly highlighted in red text. Below is a summary of the key modifications and additions:
- Clarified that SSGNN performs global context filtering rather than set-to-set filtering. (Reviewers 2, 3)
- Included visualizations in Appendix Section F: Visual Insights of the Multi-Headed Decoder, showcasing the impact of decoder heads on SSGNN’s performance. (Reviewer 2)
- Added precomputation time for SSGNN and training time per epoch (in milliseconds) comparisons in Appendix Sections G.1 and G.2. (Reviewers 1, 3, 4)
- Detailed the impact of ReGA and mean-shift operations through experimental insights in Appendix Sections G.4 and G.5, respectively. (Reviewers 2, 3, 4)
- Presented experiments regarding eigen corrections for both Specformer and SSGNN in Appendix Section G.6. (Reviewer 3)
- Studied the effect of Transformers in Specformer experimentally in Appendix Section G.3. (Reviewer 4)
- Added additional comparisons against baselines like SGC and SSGC in Appendix Section G.7. (Reviewer 1)
- Corrected repetitive symbols and clarified incomplete sentences such as “* means…”. (Reviewers 2, 4)

We have updated our manuscript based on the feedback and responded to each reviewer’s specific comments and questions.

We sincerely thank the reviewers for their valuable suggestions. Every effort has been made to address all concerns comprehensively.

We kindly request the reviewers to review the revisions and reconsider their evaluations.

Thank you for your time and consideration.

Sincerely,

The Authors

---

### Meta-Review · Area_Chair_x5gZ · 2024-12-18

**Metareview:**

In this submission, the authors proposed a new member of spectral GNNs with advantages in computational efficiency. However, some questions are unresolved:

1) The experimental part is not solid enough. Although the authors provide more analytic experimental results, the advantage of the proposed method compared with the existing GNNs and graph-oriented Transformer models is not significant or consistent.

2) As a kind of graph spectral filtering-based method, the representation power of the proposed model is not analyzed in details. For example, whether the proposed method can represent arbitrary graph filters is unknown.

3) As the authors claim, the key technical contribution of this submission is the ReGA module. However, the ablation studies provided by the authors imply that the other mechanisms, such as re-centering and eigen-correction, significantly impact the model performance.

4) The writing of the proposed submission is unsatisfying. There still are some obvious typos in the revised paper, and the organization of the proposed method can be improved.

Overall, the motivation and analytic part of the proposed method needs to be enhanced, and the submission requires a next-round review.

**Additional Comments On Reviewer Discussion:**

Two of the four reviewers interacted with the authors and decided to maintain their scores finally. AC requires more comments in the discussion-decision phase but did not get feedback. After reading the submission, the comments, and the rebuttals, AC has decided to reject this work.

---

### Decision · Program_Chairs · 2025-01-22

Reject